# Non-targeted N-glycome profiling reveals multiple layers of organ-specific diversity in mice

Johannes Helm[1,9], Stefan Mereiter[2,3,9], Tiago Oliveira[2,3], Anna Gattinger[3,4], David M. Markovitz [5], Josef M. Penninger [2,3,6,7], Friedrich Altmann [1] & Johannes Stadlmann [1,8]

N-glycosylation is one of the most common protein modifications in eukaryotes, with immense importance at the molecular, cellular, and organismal level. Accurate and reliable N-glycan analysis is essential to obtain a systemswide understanding of fundamental biological processes. Due to the structural complexity of glycans, their analysis is still highly challenging. Here we make publicly available a consistent N-glycome dataset of 20 different mouse tissues and demonstrate a multimodal data analysis workflow that allows for unprecedented depth and coverage of N-glycome features. This highly scalable, LC-MS/MS data-driven method integrates the automated identification of N-glycan spectra, the application of non-targeted N-glycome profiling strategies and the isomer-sensitive analysis of glycan structures. Our delineation of critical sub-structural determinants and glycan isomers across the mouse N-glycome uncovered tissue-specific glycosylation patterns, the expression of non-canonical N-glycan structures and highlights multiple layers of N-glycome complexity that derive from organ-specific regulations of glycobiological pathways.

Protein glycosylation, the covalent attachment of simple or complex sugar structures to amino-acid side chains of polypeptides, affects virtually all aspects of biology. Over 50% of all human proteins are subject to post-translational modifications by glycans[1], which alter their functions in fundamental biological processes, such as cell adhesion, signal transduction, intracellular trafficking, essential immune functions[2] or host-pathogen interactions[3].

In the process of mammalian protein glycosylation, glycan structures are co- or post-translationally linked to nascent polypeptide chains, which then become extensively processed along the secretory pathway, throughout the Golgi-network to the cell surface. In protein N-glycosylation, arguably the currently best-understood form of protein glycosylation, a large, highly conserved N-glycan precursor structure is *en bloc* transferred and covalently linked to the side chains of specific asparagine-residues within the endoplasmic reticulum (ER). Subsequent N-glycan processing is a non-template-driven process that critically depends on the coordinated action of dedicated enzymes, a highly specific set of glycosyltransferases and glycosidases[1].

[1]Institute of Biochemistry, Department of Chemistry, University of Natural Resources and Life Sciences (BOKU), Muthgasse 18, Vienna, Austria. [2]Eric Kandel Institute, Department of Laboratory Medicine, Medical University of Vienna, Spitalgasse 23, Vienna, Austria. [3]Institute of Molecular Biotechnology of the Austrian Academy of Sciences (IMBA), Vienna BioCenter (VBC), Dr. Bohr-Gasse 3, Vienna, Austria. [4]Bioinformatics Research Group, University of Applied Sciences Upper Austria, Softwarepark 11, Hagenberg, Austria. [5]Division of Infectious Diseases, Department of Internal Medicine, and the Programs in Immunology, Cellular and Molecular Biology, and Cancer Biology, University of Michigan, Ann Arbor, MI, USA. [6]Department of Medical Genetics, Life Sciences Institute, University of British Columbia, Vancouver Campus, 2350 Health Sciences Mall, Vancouver, BC, Canada. [7]Helmholtz Centre for Infection Research, Braunschweig, Germany. [8]BOKU Core Facility Mass Spectrometry, University of Natural Resources and Life Sciences (BOKU), Muthgasse 18, Vienna, Austria. [9]These authors contributed equally: Johannes Helm, Stefan Mereiter. ✉e-mail: j.stadlmann@boku.ac.at

Perturbations in any of these tightly interconnected and subtly tuned cellular processes hold the potential to result in differentially processed glycoproteins that exhibit aberrant glycan structures.

Owing to its close resemblance to human physiology, its short generation time, and its visible phenotypic variants, the mouse (*Mus musculus*) represents the most common mammalian model organism to study fundamental biological processes, including glycosylation[4]. Nevertheless, differences between human and murine glycosylation have been described. Most importantly, humans lack the Gal-α1,3-Gal epitope (due to an inactivation mutation in the α1,3-galactosyl transferase (*Ggta1*)[5]) and the biosynthetic capabilities for the generation of N-glycolylneuraminic acid (Neu5Gc) due to a mutation in the CMP-NeuAc hydroxylase[6] (*Cmah*). Further differences between human and murine glycosylation have recently been reported for the brain[7] and immune cells[8].

The comprehensive analysis of protein-linked N-glycan structures is a highly challenging task. Due to compositional, structural, positional, and anomeric isomers[9], this analytical challenge has long been tackled using a variety of glycomics techniques, which all profile different aspects of glycosylation and all come with specific analytical advantages and disadvantages. Importantly however, the use of this wide range of different methodologies, including lectin-arrays[10], MALDI-TOF mass-spectrometry (MS) analysis of native[11], permethylated[12] or otherwise derivatized glycans[13,14], fluorescence labeling of glycans followed by either reversed- or normal-phase HPLC analysis[15,16], various LC-MS and LC-tandem MS (MS/MS) analysis workflows[17,18], causes low comparability between studies.

Generation of glycobiological ground truth was pioneered by the Consortium of Functional Glycomics glycan profiling initiative (CFG, www.functionalglycomics.org) who generated, annotated, and made publicly available the hitherto most comprehensive, consistent, and coherent mammalian glycome datasets. From 2001 to 2011, the CFG Analytical Glycotechnology Core analyzed the compositions of N- and O-linked glycans from 11 human and 16 murine tissues, primarily using MALDI-TOF MS analysis. While the CFG's analytical activities have come to an end, their seminal glycan profiling datasets still represent a vital and fundamental resource to the field of mammalian glycobiology. More recently, new glyco-analytical developments and the paucity in modern, comprehensive N-glycome datasets prompted several studies aiming at the systemic profiling of tissue-specific N-glycosylation patterns using MALDI-TOF MS[19–24]. Despite the remarkable compositional variations that were detected by such (single-stage) MS approaches, these methods intrinsically fall short in capturing the unique level of N-glycan structure micro-heterogeneity. Instead, glycan structures are tentatively deduced from composition and based on pre-established knowledge of glyco-biosynthetic pathways.

In this work, we map the N-glycomes of 20 mouse tissues by performing PGC-LC-MS/MS (porous graphitic carbon liquid-chromatography coupled to a tandem mass-spectrometer). We use PGC-LC to chromatographically separate closely related N-glycan structure isomers and an Orbitrap mass-analyzer for high resolution MS/MS data-acquisition. Conventionally, the analysis of such LC-MS(/MS)-N-glycome datasets is based on the targeted, selective extraction for known or anticipated glycan-compositions and -masses[20,22,25]. Subsequent glycan-structure assignment incorporates additional information on pre-established, relative chromatographic retention times[9,26,27], complementary MS/MS data and expert knowledge on glycan biosynthetic pathways. Despite the breath of glycan-information that can be retrieved by such -largely manual-approaches[9,25,27–30], they are hardly scalable to the size of the present dataset. Instead, we develop an automated, non-targeted and scalable MS/MS-centric N-glycomics data-analysis workflow, largely independent of prior glycobiological knowledge and anticipated glycan-compositions. We showcase this approach by characterizing 20 different mouse tissues, providing a consistent and complete N-glycome atlas. Our analyses reveal tissue specific N-glycome signatures and glycan-structural features, highlighting organ-intrinsic regulations of glycobiological pathways.

## Results

### Precursor-independent N-glycome profiling of 20 mouse tissues by MS/MS spectral counting

To generate a consistent N-glycome dataset, we collected tissues, as well as serum, in duplicates from age-matched C57BL/6J mice, and processed all samples using identical protein extraction, enzymatic N-glycan release (i.e. PNGase F), chemical reduction, and glycan clean-up protocols. All samples were analyzed using porous graphitic carbon liquid-chromatography (PGC-LC) coupled to a high-resolution tandem (MS/MS) mass-spectrometer (i.e. Orbitrap Exploris 480).

To survey this data, we first assessed the MS/MS data for the occurrence of diagnostic, N-glycan-derived fragment ions, independent of intact glycan precursor mass information. For this, we automatically extracted all MS/MS spectra and retained only those that contained N-glycan-specific fragment ions (i.e. 224.1 amu, diagnostic for reduced N-acetylhexosamine). From this data, we generated semi-quantitative information on the MS/MS level by implementing a spectral counting approach, similar to methods used in the fields of proteomics[31,32] More specifically, we developed a binary MS/MS spectrum counting approach, that is precursor-mass and -intensity independent and exclusively based on the detection (or complete lack) of N-glycan specific fragment ions. Using this approach, a first breakdown of our dataset confirmed that, overall, more than 40 percent of all MS/MS spectra generated in this study (i.e. 220,506 out of 509,283 MS/MS spectra) contained information on our target analytes, namely chemically reduced glycans. Manual inspection of rejected MS/MS spectra (i.e. MS/MS spectra that did not exhibit the 224.1 amu fragment ion) showed that a big share of these spectra did not contain any other known glycan fragments either and were thus deemed to having been derived from non-glycan contaminants (and therefore correctly filtered out by our script).

Importantly, however, we found considerable differences in the relative proportion of N-glycan-derived MS/MS spectra between tissues, ranging from approx. 20% for ileum to more than 80% for seminal vesicle (Supplementary Fig. 1A). This tissue-dependent variability in the dynamic range and structural complexity prompted further investigation into tissue-specific N-glycome features such as sialylation or fucosylation, independent of intact glycan precursor information.

Sialylation is essential to mammalian development[33] and results from the attachment of either N-acetyl-neuraminic acid (Neu5Ac) or N-glycolyl-neuraminic acid (Neu5Gc). To dissect this important compositional heterogeneity of N-glycans, we extended our MS/MS data-filtering criteria to sialylation-specific diagnostic fragment ions (i.e. reduced HexNAc fragment ion mass 224.1 amu and Neu5Ac fragment ion mass 292.1 amu and/or Neu5Gc fragment ion mass 308.1 amu) and counted the associated MS/MS spectra. To ensure that only spectra derived from reduced N-glycans are included, all spectra without the 224.1 amu fragment ion were excluded before the counting step. This simplistic approach revealed substantial differences in the number of Neu5Ac- and/or Neu5Gc-containing spectra across tissues, ranging from approximately 80% in serum, lung, and heart to only about 18% in the seminal vesicle (Fig. 1, Supplementary Table 1). Most tissues exhibited comparable levels for both sialic acid variants (Supplementary Fig. 1B) (mean ratio Neu5Gc:Neu5Ac = 1.5), except for serum (Neu5Gc:Neu5Ac = 11.2), colon (Neu5Gc:Neu5Ac = 0.3), and brain (Neu5Gc:Neu5Ac = 0.1), which showed notably different ratios. This data corroborated previous reports on the murine brain N-sialome being vastly dominated by Neu5Ac and to only contain trace amounts of Neu5Gc[28,34,35]. Overall, heart, kidney, liver, lung, mammary gland, serum, skin, spleen, testis, and thymus showed higher numbers of sialic acid-containing spectra compared to the mean value across all

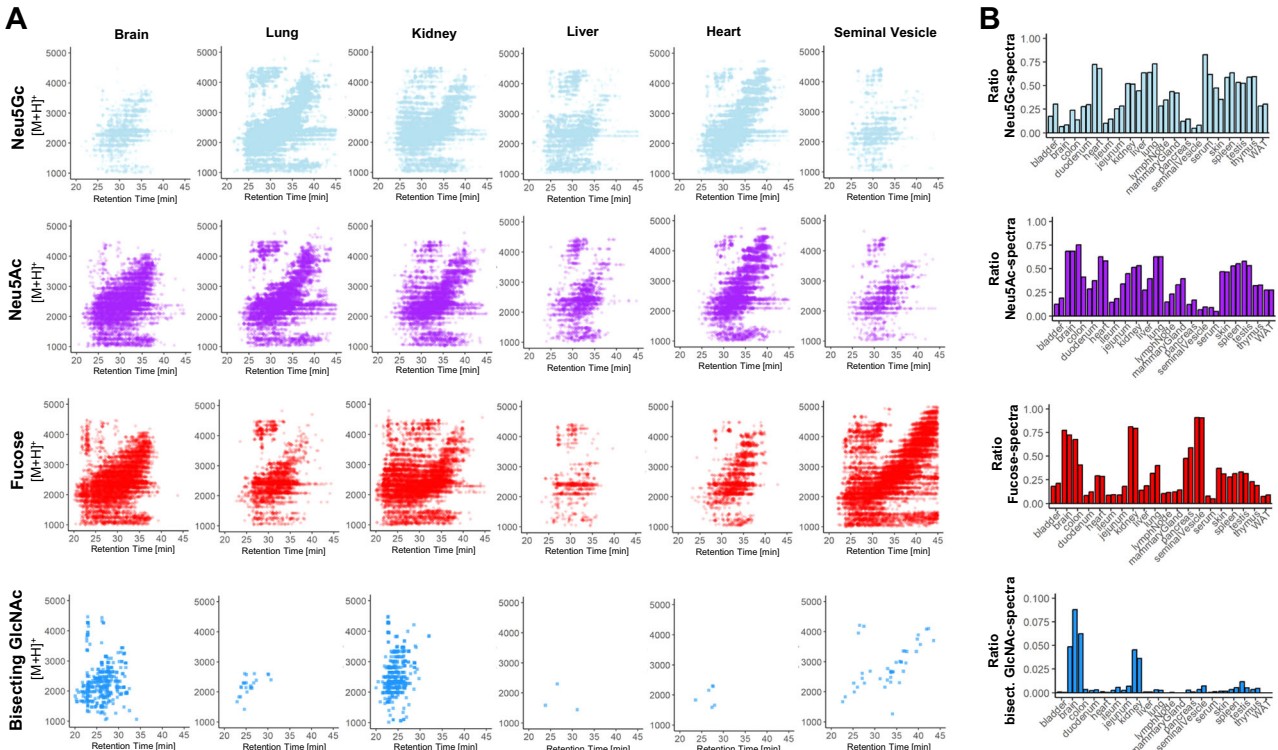

**Fig. 1 | Precursor-independent MS/MS-based N-glycome profiling. A** top panel: Biosynthetic pathway leading to complex-type and hybrid-type N-glycans. lower panel: Precursor mass over chromatographic retention time plots MS/MS spectra. Each data point represents a single N-glycan derived MS/MS spectrum (i.e containing diagnostic fragment ions at 224.1 amu and one of the following signals: 274.1 amu for Neu5Ac-sialylation, 290.1 amu for Neu5Gc-sialylation, 512.2 amu for fucosylation, and 792.3 amu for bisecting GlcNAc). **B** Relative spectral count. Data were normalized to the total number of N-glycan-derived MS/MS spectra (i.e. containing 224.1 amu) in the respective tissues. All tissues were analyzed in duplicates. Numeric results of the spectral counting are provided in Supplementary Table 1. WAT - white adipose tissue.

tissues, while bladder, brain, colon, duodenum, ileum, jejunum, lymph node, seminal vesicle, pancreas, and white adipose tissue exhibited lower numbers. (Supplementary Figs. 2A–C).

Similar to sialylation, fucosylation is a vital, developmentally controlled modification of N-glycans, which has been implicated in numerous cell-cell interactions[36] and massively expands the structural heterogeneity of the N-glycome. Fucose can be linked to the most proximal core-GlcNAc residue, originally connected to the protein (i.e. core-fucose). Additionally, fucose residues have been found linked to either galactoses or N-acetylhexosamines in multiple positions of the distal parts of N-glycan antenna (i.e. distal fucosylation). Interestingly, distal fucosylation also encodes a series of essential, immune-reactive glycan-epitope isomers, comprising one (e.g. Lewis X, sialyl Lewis X, Lewis A, sialyl Lewis A, blood-group H type 1 and type 2) or more fucose residues (e.g. Lewis Y, Lewis B), in humans. Fucosyltransferase 3 (FUT3), the enzyme which catalyzes the addition of the Lewis A and Lewis B epitope in humans was found to be a pseudogene in mice. The murine system is thus believed to lack these two important fuco-epitopes[37,38]. Unfortunately, most of these fucosylated N-glycan isomers cannot be resolved by MS/MS alone[39–41]. As a consequence, our precursor independent spectral counting approach, which is based on fragment ions that are indicative[39–41] of fucosylated N-glycans (i.e. one fucose linked to one HexNAc and to one hexose; fragment ion mass = 512.2 amu), merely reflects on the combined expression patterns of core fucose, Lewis X, sialyl Lewis X, Lewis Y, blood-group H (bgH) type 1 and type 2, but not Lewis A or Lewis B, in murine tissues. Our analysis showed that fucosylated N-glycans were indeed present in all tissues, with exceptionally high levels in seminal vesical (85%)[42], kidney (65%), and brain (50%) (Fig. 1, Supplementary Fig. 2E). This suggested generally high expression levels of fucosyltransferases in these tissues, such as *Fut9*, which is highly expressed in kidney and brain[43], *Fut2* and

*Fut4*, which are highly expressed in colon epithelial cells, or *Fut8*, which is globally expressed in mouse[44].

Another distinctive structural feature of N-glycans is bisecting N-acetylglucosamine (GlcNAc). Bisecting GlcNAc residues are β1,4-linked to the core β-mannose residue by MGAT3 and have been reported vital to fetal development[45], immunity, and cell adhesion[46]. Expression of this critical structural modification of N-glycans was profiled across all tissues, based on the diagnostic fragment ion of mass 792.3 amu (i.e. one reduced GlcNAc-residue, two GlcNAc-residues, and one hexose residue). As expected, we found bisecting GlcNAc expression levels to be highly tissue-specific, with the highest expression levels in the brain (4%), kidney (2.5%), and colon (2%) (Fig. 1, Supplementary Fig. 2D), all in good agreement with mouse *Mgat3* gene expression data[47]. The marked differences between the colon duplicates may be a result of different levels of residual mucus, food stuff or fecal matter in the colonic sections analyzed.

Moreover, an extensive screening of all tissues was conducted to identify fragment ions indicative of the Sda antigen. This antigen is characterized by a Neu5Ac residue α2,3-linked and a GalNAc residue β1,4-linked to the galactose within a LacNAc motif (860.3 amu)[48]. Intriguingly, this specific structural modification was predominantly expressed in the colon (9%), followed by the jejunum (3.5%) and duodenum (1%) (Supplementary Fig. 1C). Notably, the ileum exhibited only minimal expression, accounting for less than 0.25% of all N-glycan associated MS/MS spectra. In previous studies, the Sda antigen has been identified in the colon of healthy humans, on the Tamm-Horsfall glycoprotein of Sda+ individuals, and in the serum of patients with gastric cancer[48]. In humans, the biosynthesis of this antigen is driven by the *B4GALNT2* gene, which orthologs' (*B4Galnt2*) expression indeed appears to be restricted to the intestinal organs in mice[44].

Although merely providing rough estimates on the relative abundance of specific N-glycan features, our MS/MS-based N-glycome profiling approach presents an easy, fast, and robust way to automatically screen individual samples for relevant N-glycome modifications.

## Automated MS/MS-data-driven reconstruction of the mouse N-glycome

To determine the N-glycan precursors contributing to the observed N-glycome signatures and to reconstruct quantitative N-glycosylation patterns at the precursor level, we expanded our binary spectral counting/filtering approach and implemented a more sophisticated data-aggregation workflow. This large-scale data-analysis pipeline leverages fragmentation (i.e. MS/MS) data for glycan precursor identification, extracts quantitative information on the precursor (i.e. MS) level, and efficiently reduces PGC-LC-MS data complexity.

The automated extraction of precursor mass information from LC-MS/MS data is often complicated by the imperfection of mass-spectrometric data, due to e.g. incorrect mono-isotopic peak-picking or charge-state assignments by the mass-spectrometer, unintended precursor ion co-isolation, in-source fragmentation, or excessive adduct-ion formation. To address these challenges, first, raw glycan-fragment data were refined (i.e. mono-isotopic peak picking and charge state assignment re-evaluation) and converted into the generic.mgf file-format using the proteomics software PEAKS[49]. From this, to automatically identify MS/MS spectra that derived from reduced N-glycan precursors and to stringently control for unintended precursor ion co-isolation events, we calculated spectrum-specific Score-values (i.e. SNOG-score, Supplementary Note 1) from the intensity of an N- (and O-) Glycan-specific fragment ion (i.e. oxonium ion of the reduced-end monosaccharide GlcNAc; 224.1 amu), using custom code (Supplementary Software). In contrast to our initial, binary spectral filtering approach, SNOG-scores incorporate critical information on fragment ion intensities. While this makes SNOG-score-based MS/MS data filtering generally more robust against e.g. co-isolation events, it is important to note that it also renders SNOG-score values dependent on the actual experimental MS/MS acquisition parameters (e.g. collision energy settings). For our dataset (i.e. generated using stepped HCD collision energies 20, 25, and 30%), we empirically determined that MS/MS spectra with SNOG-scores greater than 0.03 derived from actual N-glycan precursors (Supplementary Figs. 3–5). Other experimental parameter settings may require different SNOG-score value thresholds for effective data-filtering.

Next, we consolidated the raw LC-MS data. For this, we charge-deconvoluted and deisotoped all individual MS spectra (using DeCon2[50], Supplementary Data 1), assigned all precursor mass and intensity values to mass bins (range of 1000–4000 Da; bin-width = ±0.05 amu), and summed their values across the entire chromatographic time-range, using custom code. This simple transformation of time-resolved LC-MS data into two-dimensional precursor mass-to-intensity arrays efficiently reduced the dimensionality of our dataset. Additionally, these data arrays allowed for the automated construction of tissue-specific quantitative histograms (Fig. 2A, Supplementary Data 2 and 3), that closely resemble MALDI-TOF MS spectra and thus provide a convenient data visualization format that integrates seamlessly with current glycome data repositories (e.g. CFG, www.functionalglycomics.org; Fig. 2A).

Eventually, precursor mass information of SNOG-scored MS/MS spectra was aligned with our two-dimensional MS precursor mass-to-intensity arrays, and only MS signals of true N-glycan precursors (i.e. at least 1 MS/MS spectrum of SNOG score greater than 0.03) above a cumulative intensity threshold of 5E + 6 were retained.

The number of the automatically identified glycan precursor masses greatly varied between tissues. For example, while SNOG-filtering reduced the total number of glycan-derived precursor mass-bins by approx. 30% in brain, it removed approx. 75% of all precursor mass-bins in the liver (Supplementary Fig. 4B). This, again, highlighted important tissue-dependent differences in the dynamic range and the structural complexity of N-glycomes, and suggested sample-specific background signals at the precursor level. Manual inspection of MS/MS spectra of low-scoring (i.e. rejected) mass-bins confirmed, for example, exceptionally high levels of non-N-glycan signals that derived from hexose-oligomers (e.g. dextrans or maltodextrins degradation products of glycogen[51]) in the liver, which were efficiently removed by our SNOG-score based filtering approach (Supplementary Fig. 5).

The correlation and cluster analysis of our SNOG-filtered dataset (Fig. 3) revealed distinctive tissue-specific clustering patterns. Notably, we found clustering even among distantly related mouse tissues, such as the exocrine organs, i.e. seminal vesicles and pancreas, clustering with the brain, or the central organs of the immune system, spleen, and thymus, coalescing with mammary glands, for which we observed convergent N-glycan signatures. Intriguingly, while the spleen and thymus share a cluster, lymph nodes exhibit a glycosylation pattern more akin to white adipose tissue, potentially influenced by spatial/histological proximity.

A distinctive glycosylation pattern set apart the tissues of the small intestine, particularly the jejunum and duodenum, constituting a distinct cluster separate from the ileum and colon. Conversely, the ileum and colon formed a cluster with the bladder and skin, suggesting a potential association rooted in shared epithelial glycosylation patterns. Additionally, our observations extend to the highly blood-perfused organs, where the liver and lung share a cluster with serum, while the heart, kidney, and testis form another distinct cluster. The measured samples were derived from different animals and were individually processed. The yet high degree of similarity across all tissue pairs evidences a high technical reproducibility of our analyses (Fig. 3). This clustering pattern provides insights into the glycosylation variations within these organ groups, potentially reflecting their functional relationships or physiological roles.

## Dissecting the mouse N-glycome based on sub-structural determinants

To systematically query and stratify N-glycan precursors information based on fragmentation data, in the next step, we further extended our SNOG-scoring scheme by additional N-glycan specific fragment ions (i.e. eSNOG). MS/MS spectrum-specific eSNOG-scores were thus calculated from the relative intensities of sub-structure specific diagnostic fragment ions and -again- empirically determined, sub-structure specific eSNOG cut-off values were used for down-stream data-filtering (Supplementary Fig. 6, Supplementary Note 2), efficiently limiting the impact of potential gas-phase rearrangements[52]. Importantly, this versatile data-filtering approach allows for the automated stratification of our reconstructed N-glycome data by integrating MS/MS information to even distinguish between certain isobaric N-glycan structures (e.g. Neu5Ac and Neu5Gc-sialylated structures, alpha-galactosylated structures and non-alpha-galactosylated structures, or core fucosylated and antenna-fucosylated N-glycans) (Fig. 4).

First, we used eSNOG values to automatically stratify the N-glycome of mouse kidney (Fig. 4.) based on signals that contained: (I) distal-fucosylated (dHex-Hex-HexNAc), (II) Neu5Gc-sialylated, (III) NeuAc5-sialylated, (IV) alpha-galactosylated (Hex-Hex-HexNAc), (V) oligomannosidic N-glycans and (VI) compositions which do not fall into any of these categories (i.e. predominantly undecorated or unusually decorated structures). Of note, even though an additional sub-categorization would be possible (the Neu5Ac-sialylated category can e.g. be further split up into Neu5Ac linked to GalNAc-residues, Neu5Ac linked to GlcNAc-residues, poly Neu5Ac, and O-acetylated Neu5Ac) here, we focus our analysis on diagnostic fragment ions that are common to all Neu5Ac-containing N-glycan sub-categories by considering only the Neu5Ac-residue with and without water loss (292.1

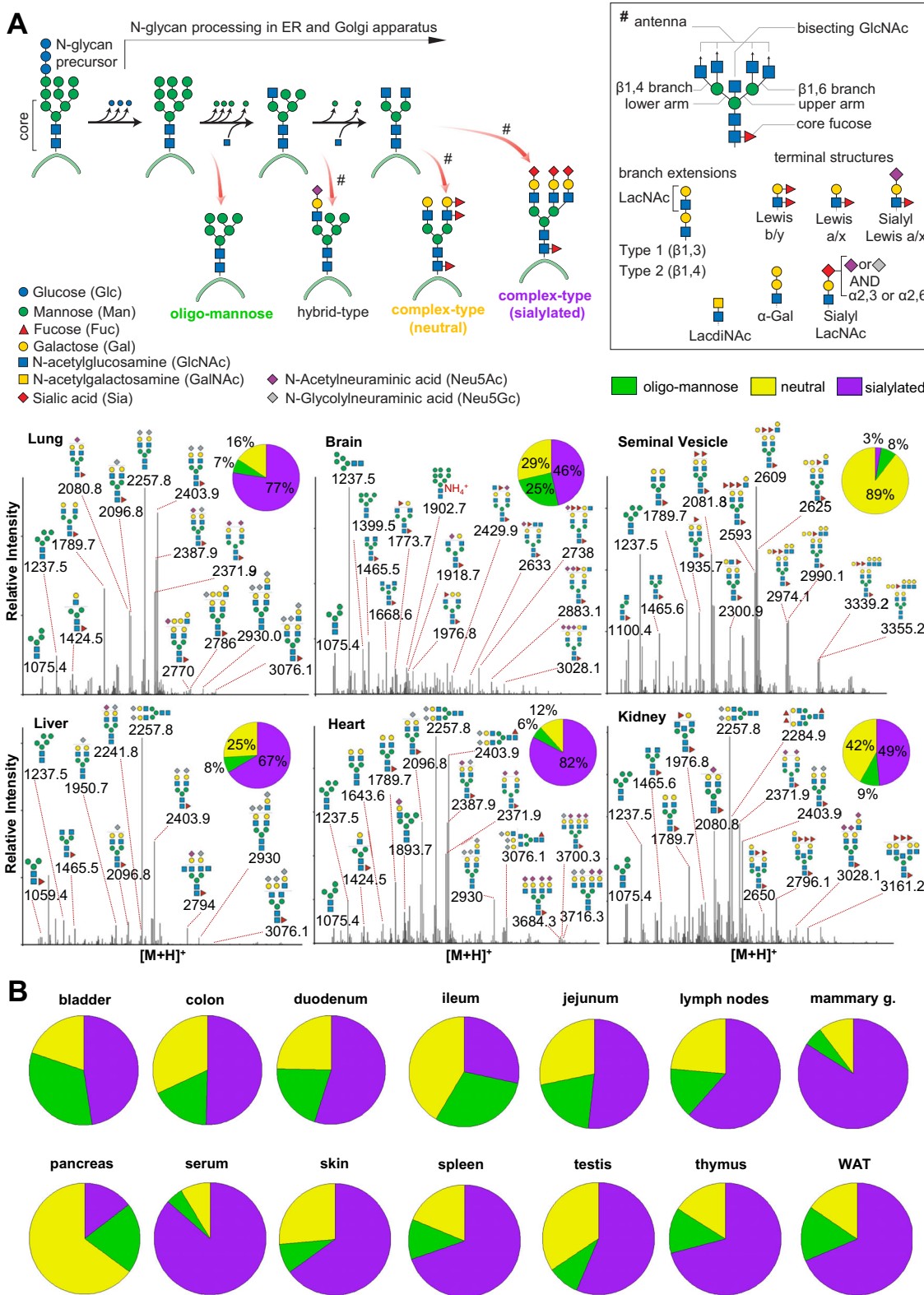

**Fig. 2 | Comparative N-glycome analysis. A** Semi-quantitative histograms of SNOG-filtered LC-MS data of selected tissues. LC-MS data were charge-deconvoluted, deisotoped, and summed across the chromatographic time-range using custom code. Precursor mass-to-intensity arrays are shown in the range of 1000–4000 Da (bin-width = ±0.05 amu). Representative, tissue-specific N-glycan compositions are indicated by their precursor mass (i.e. [M + H]⁺ in amu) and tentative structure assignments using the Symbol Nomenclature for Glycans (SNFG).

Signals below a cumulative intensity threshold of 5E + 6 and a SNOG-score of 0.03 were removed. Inset pie charts show the N-glycan compositional fractions of the respective N-glycome (green: oligomannosidic- (OM), purple: sialylated (including both Neu5Ac and Neu5Gc-sialylated glycans), yellow: undecorated/neutral N-glycans). **B** N-glycan compositions of tissues not included in Panel A. WAT- white adipose tissue, Lymph N - lymph nodes, Mammary G. - mammary gland.

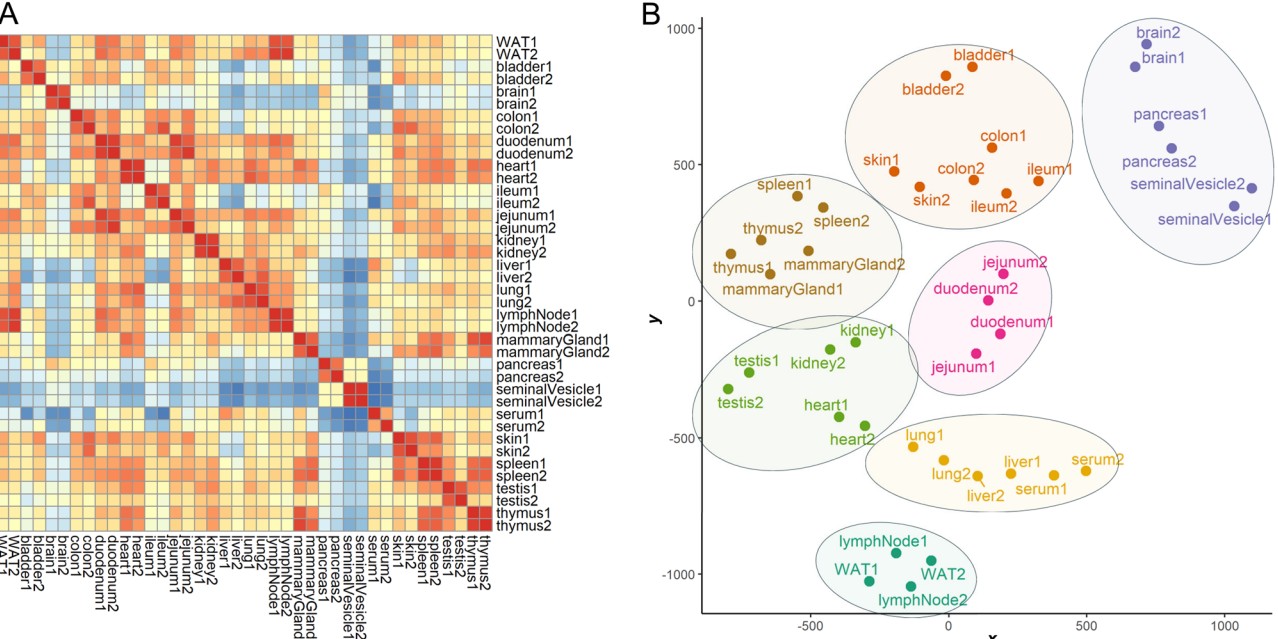

**Fig. 3 | Correlation and cluster analysis of SNOG-filtered LC-MS data.**
**A** Correlation heatmap was calculated using Pearson´s correlation coefficients.
**B** t-SNE plot. Hierarchical clustering was based on Euclidean distances of the calculated t-SNE values. Clustering was calculated by Ward´s method ("ward.D2").

Prior to analysis transformed LC-MS data were SNOG-filtered and intensity values were normalized to TIC. Signals below a cumulative intensity threshold of 5E + 6 and a SNOG-score of 0.03 were removed. WAT - white adipose tissue.

and 274.1 Da, respectively). A list with the diagnostic fragment ions used in this study, as well as the respective thresholds applied, can be found in Supplementary Table 2.

In total, our eSNOG approach automatically uncovered 227 fucose-, 180 Neu5Gc-, 207 Neu5Ac-, 31 alpha-Gal-containing N-glycan precursor mass-bins, as well as 82 N-glycan precursors that do not fall into any of the six categories. When accumulating the MS signal intensities of all N-glycan precursors that fell into these six different categories, distal fucosylated N-glycans made up ~34% of the total ion current (TIC), Neu5Gc-sialylated mass-bins ~30%, Neu5Ac-sialylated mass-bins ~23%, alpha-Gal containing mass-bins ~4%, and oligomannose-associated mass-bins ~8%. Glycan compositions that did not conform to any of these categories (i.e. undecorated structures), represented ~16% of the TIC in kidneys.

Manual inspection of the stratified kidney N-glycome dataset (Fig. 4A) confirmed that our approach accurately classified N-glycan precursor mass-bins. For example, based on their precursor masses, biantennary, core-fucosylated and di-sialylated N-glycans were correctly assigned to either contain only Neu5Ac (2371.9 amu, $Hex_5HexNAc_4Fuc_1Neu5Ac_2$), only Neu5Gc (2403.9 amu, $Hex_5HexNAc_4Fuc_1Neu5Gc_2$), or one Neu5Ac as well as one Neu5Gc-residue (2387.9 amu, $Hex_5HexNAc_4Fuc_1Neu5Ac_1Neu5Gc_1$) (Fig. 4A). Similarly, the mass-bins of 2257.8 amu and 2113.8 amu were correctly assigned to represent Neu5Gc sialylated ($Hex_5HexNAc_4Neu5Gc_2$, Fig. 4B) and alpha-galactosylated ($Hex_7HexNAc_4Fuc_1$) N-glycan precursor compositions, respectively. Furthermore, the mass-bin of 2039.7 amu was classified to contain Neu5Ac, which allowed us to accurately define its composition as $Hex_6HexNAc_3Fuc_1Neu5Ac_1$, corresponding to a Neu5Ac-sialylated hybrid-type N-glycan. The Neu5Gc-capped analog of the same glycan at 2055.7 amu (i.e. $Hex_6HexNAc_3Fuc_1Neu5Gc_1$) was correctly classified as Neu5Gc-containing N-glycan. The most abundant fucosylated N-glycan precursor in mouse kidneys corresponded to a presumably bisected, bi-antennary N-glycan with three fucose residues (2284.9 amu) (Fig. 4A). Other fucosylated precursor masses corresponded to tri- or tetra-antennary extensions of this N-glycan composition, with or without bisecting GlcNAc and with 3 to 5 fucose residues

(e.g. $Hex_6HexNAc_5Fuc_{3/4}$, $Hex_6HexNAc_6Fuc_{3/4}$, $Hex_7HexNAc_6Fuc_{3/4/5}$, $Hex_7HexNAc_7Fuc_{3/4/5}$, Fig. 4B), confirming previous studies[53,54]. Abundant N-glycan precursor masses that were classified as undecorated included the mass of 1789.7 amu, which corresponded to the composition $Hex_5HexNAc_4Fuc_1$, as well as smaller, truncated N-glycans, including the compositions $Hex_3HexNAc_2Fuc_1$ and $Hex_3HexNAc_3Fuc_1$ (Fig. 4A), which are indeed not modified by distal fucose or sialic acid residues.

Capitalizing on our automated data-analysis workflows, we next stratified all other N-glycome datasets into the same six N-glycan categories and compared their relative abundances across the 20 murine tissues analyzed (Fig. 4C). This comparative analysis allowed us to further dissect the structural complexity of the mouse N-glycome at the precursor level and revealed a remarkable diversity in glycosylation patterns across tissues.

From our initial precursor-independent profiling analyses we found that serum was largely dominated by Neu5Gc-bearing N-glycans, representing up to 96% of the TIC[55]. Additionally, our N-glycan precursor-informed data now revealed that this important trait of the murine serum N-glycome was essentially derived from only two biantennary, non-fucosylated N-glycans ($Hex_5HexNAc_4Neu5Gc_1$ and $Hex_5HexNAc_4Neu5Gc_2$), totaling ~57% of all its Neu5Gc-containing structures. The sialylated fraction of the brain N-glycome, on the other hand, consisted almost entirely of Neu5Ac-decorated N-glycan species, in good agreement with previous studies[25,56]. Based on our eSNOG-stratified precursor information, the N-sialome of the brain presented as highly diverse, comprising truncated (e.g. $Hex_4HexNAc_3Fuc_1Neu5Ac_1$), hybrid-type (e.g. $Hex_6HexNAc_3Fuc_1Neu5Ac_1$), bisecting GlcNAc-containing (e.g. $Hex_5HexNAc_5Fuc_1Neu5Ac_2$) and highly complex tetra-antennary N-glycans (e.g. $Hex_7HexNAc_6Fuc_3Neu5Ac_2$). Additionally, we found approx. 50% of the sialylated N-glycan species in brain to also carry distal fucoses. From this data, we calculated that almost 50% of the brain N-glycome is sialylated. Of note, the degree of sialylation reported for the brain varies vastly between studies, with previous estimates ranging from only ~3%[25], over 20%[29], or up to ~40% of sialylation[57], depending on the methodology used.

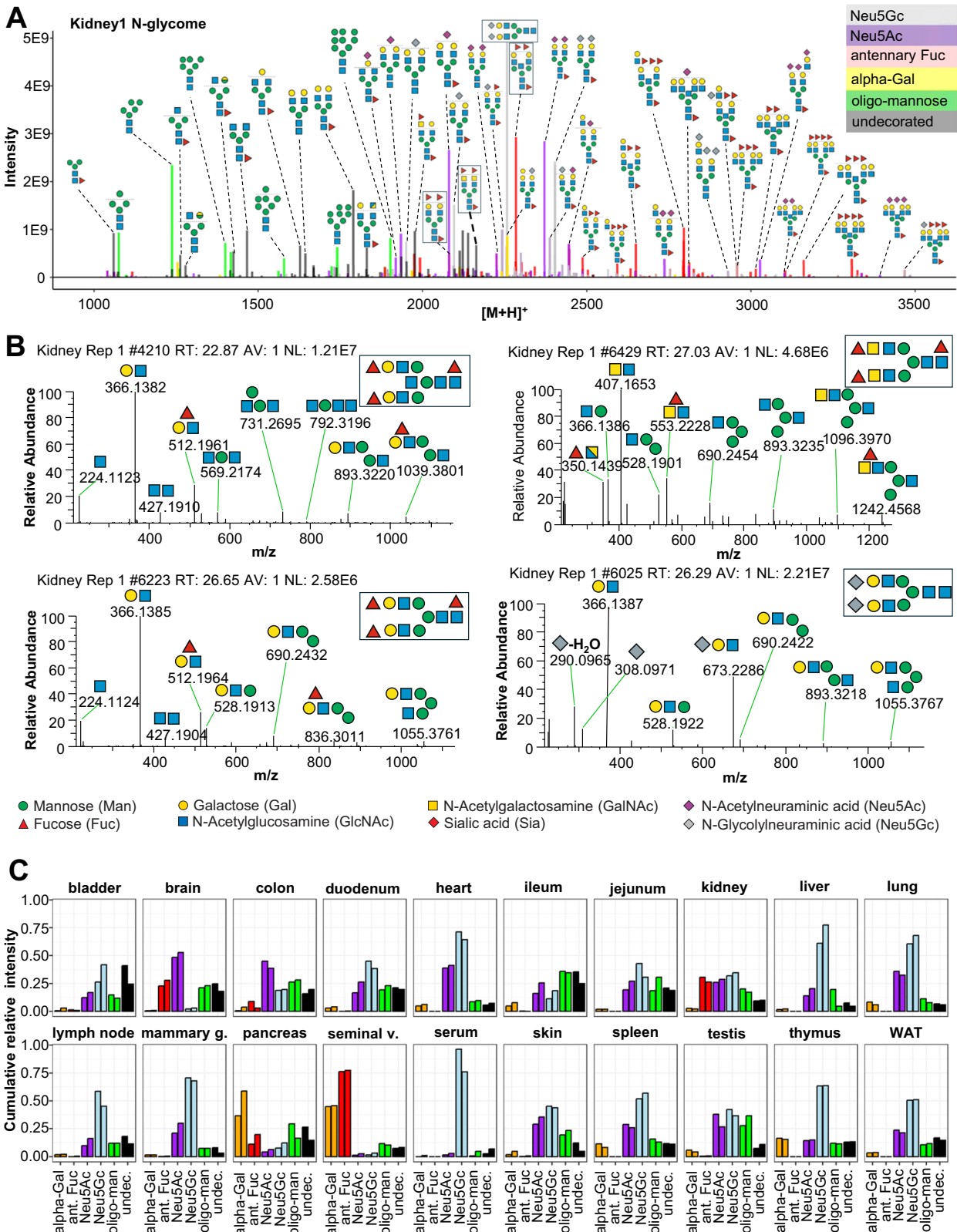

**Fig. 4 | Sub-structural stratification of the mouse N-glycome. A** eSNOG-based classification of the kidney N-glycome. LC-MS signals were automatically categorized based on diagnostic fragment ions using sub-structure specific eSNOG cut-off values and classified as: distal-fucosylated (red), Neu5Gc-sialylated (light blue), NeuAc5-sialylated, (purple), alpha-galactosylated (yellow), oligomannosidic N-glycans (green), and glycan compositions that do not fall into any of the other categories (black). N-glycan cartoons depict tentative structure assignments based on composition, using the Symbol Nomenclature for Glycans (SNFG). **B** Exemplary MS/MS data of representative kidney-derived N-glycan structures. Diagnostic fragment ion compositions are indicated as cartoons, using the SNFG. **C** Relative abundance of sub-structural determinants across tissues. Both replicates for each tissue are shown. Ant. Fuc - antennary fucosylation, Oligo Man - oligomannosidic, Undecor. – undecorated (i.e. do not fall into any of the other categories).

Next to the brain, the kidney and seminal vesicle stood out by their comparably high abundance of distal fucosylated N-glycans. Different from the brain and kidney, however, seminal vesicles showed low levels of sialylation, with only ~3% of the total N-glycome being either Neu5Ac- or Neu5Gc-sialylated. Instead, the seminal vesicle N-glycome was dominated by distal fucosylated (~75%) N-glycans that were at the same time alpha-galactosylated (~45%). The most abundant precursor compositions of this type corresponded to a series of tri-antennary, core-fucosylated N-glycans with permutational additions of alpha-galactose and/or distal fucose (i.e. $Hex_6HexNAc_5Fuc_4 - Hex_7HexNAc_5Fuc_3 - Hex_8HexNAc_5Fuc_2$), suggesting that fucosylation and alpha-galactosylation on the same antenna are mutually exclusive. Similarly high levels of alpha-galactosylated N-glycans were only found in the pancreatic N-glycome (~47%), which also presented with rather low levels of sialylation (approx. 10%) and distal fucosylation (approx. 15%). In the pancreas, the most abundant alpha-galactosylated N-glycans ranged from biantennary, with or without core-fucose, with one or two alpha-galactosylated or fucosylated antennae (e.g. $Hex_6HexNAc_4Fuc_1$, $Hex_7HexNAc_4$, $Hex_6HexNAc_4Fuc_2$, $Hex_7HexNAc_4Fuc_1$), to tri- and tetra-antennary antennary N-glycans with alpha-galactosylated and/or fucosylated antenna (e.g. $Hex_7HexNAc_5Fuc_2$, $Hex_8HexNAc_5Fuc_1$, $Hex_7HexNAc_5Fuc_3$, $Hex_8HexNAc_5Fuc_2$, $Hex_9HexNAc_5Fuc_1$, $Hex_{10}HexNAc_6Fuc_2$, $Hex_{11}HexNAc_6Fuc_1$). The exceptionally high levels of alpha-Gal in the two exocrine organs, seminal vesicle and pancreas, were not observed in previous mouse studies.

Differentiating the N-glycomes of seminal vesicle and pancreas, we next screened our data for fragment ions that are diagnostic for doubly fucosylated antennae (i.e. $Hex_1HexNAc_1Fuc_2$, 658.3 amu), hence the Lewis Y-epitope. Our analysis revealed that this N-glycan motif was most specific for seminal vesicle (~1.2%) (Supplementary Fig. 7A) and barely detected in the pancreas of mice. Lewis Y-containing precursors included a series of partially core-fucosylated, bi- (e.g. $Hex_5HexNAc_4Fuc_4$, $Hex_5HexNAc_4Fuc_5$,) and multi-antennary N-glycans (e.g. $Hex_6HexNAc_5Fuc_5$, and $Hex_8HexNAc_7Fuc_6$). This unique glycosylation landscape of the seminal vesicle stands out among all tissues. Furthermore, the high expression levels of distally fucosylated N-glycans, including those with the Lewis Y epitope, correlated with gene expression data of the respective fucosyltransferases (i.e. *Fut2*, *Fut4*), as well as previous reports on the glycome of human seminal plasma[42,58]. The functional implications of these exceptionally high levels of distal fucose and alpha-galactose in exocrine organs remain to be explored.

## Deep mining the mouse N-glycome uncovers unusual structural modifications

To comprehensively annotate the results of our non-targeted data-analysis approach we used an in silico constructed mouse N-glycome database, holding an extensive list of canonical mouse N-glycans (i.e. glycoDB; 1429 unique N-glycan compositions, 960 precursor ion mass bins; Supplementary Data 3) Surprisingly, the automated annotation of our N-glycome histograms with this glycoDB highlighted several precursor masses, which, despite their consistent and reproducible detection, could not be explained by known, conventionally computed or anticipated N-glycan compositions. We hypothesized that this remarkably large population of unknown N-glycan precursor masses (i.e. approx. 15% of TIC across all samples; ranging from approx. 6% TIC in the ileum, to up to approx. 37% in liver; Supplementary Figs. 4C and D) would either result from experimental noise in the MS raw-data or represent unusual N-glycan compositions. Manual inspection of underlying MS/MS spectra uncovered and confirmed a series of unusual diagnostic fragment ions that were indicative of rare or non-canonical N-glycan structures, such as those bearing the HNK-1 epitope, sulfated HexNAc residues, doubly sialylated antenna, fucosylated LacdiNAc structures, or acetylated sialic acids (i.e. Ac-Neu5Ac and Ac-Neu5Gc), not covered by our *in-silico* N-glycome model database. Systematic mining of our dataset across all tissues using our eSNOG

approach revealed tissue-specificity for most of these non-canonical N-glycan modifications. A list of all diagnostic fragment ions and their associated thresholds used in this study can be found in Supplementary Table 2.

HNK-1 is a unique trisaccharide epitope that consists of glucuronic acid linked to galactose, which is further linked to a GlcNAc residue. Additionally, HNK-1 has been reported to exist in a sulfated and a non-sulfated form (i.e. HSO3-3GlcA-beta1,3-Gal-beta-1,4-GlcNAc or GlcA-beta-1,3-Gal-beta-1,4-GlcNAc)[59,60]. This usually unregarded modification was previously identified in brain[28,59] and kidney[60], mainly in biantennary complex type N-glycan structures. The attachment of glucuronic acid is known to be catalyzed by two, highly tissue-specific, homologous enzymes, namely GlcAT-P (*B3gat1*) in the brain and GlcAT-S (*B3gat2*) in the kidney. Intriguingly, while GlcAT-P activity is suppressed by bisecting GlcNAc residues, the activity of GlcAT-S is not affected[61]. Mining the entire mouse N-glycome for diagnostic fragments that relate to HNK-1, either sulfated or non-sulfated (i.e. SO4-HexA-Hex-HexNAc; HexA-Hex-HexNAc), confirmed that HNK-1 was indeed only expressed in kidney (~1.5%)[60] and brain (~0.5%) (Fig. 5A). The sulfated form of HNK-1, however, was exclusively found in the brain. In kidney, the most abundant HNK-1 N-glycan precursor corresponded to a triply fucosylated, tri-antennary structure, carrying a bisecting GlcNAc, with the composition $Hex_6HexNAc_6Fuc_3HexA_1$. Other HNK-1 containing N-glycans of this tissue corresponded to tri- and tetra-antennary structures with 1–3 antennary fucose-residues (e.g. $Hex_6HexNAc_6Fuc_4HexA_1$, $Hex_7HexNAc_7Fuc_2HexA_1$, $Hex_7HexNAc_7Fuc_3HexA_1$, $Hex_7HexNAc_7Fuc_4HexA_1$). It is noteworthy that, based on its composition, and corroborated by the underlying MS/MS spectra, we found that the $Hex_6HexNAc_6Fuc_4HexA_1$ N-glycan carried a fucosylated HNK-1 antenna (Supplementary Fig. 8). To the best of our knowledge, this fucosylated form of HNK-1 has never been found in a natural source and was only artificially synthesized in a previous study[62].

Sulfation is considered the most diverse glycan modification with 35 sulfotransferases involved in the process of glycan sulfation[63]. Although many of these sulfotransferases are thought to serve in decorating O-linked glycosaminoglycan chains, sulfated N-glycans have also been reported for porcine and human pancreas, with the highest levels found within the islets of Langerhans[64]. Systematic MS/MS-based screening of the different tissues for the relevant fragment ion (HexNAc-SO4 – 284 amu) revealed that almost all tissues exhibited at least low levels of N-glycan sulfation. In line with previous reports[34,64], also in our data the highest amount of HexNAc sulfation was found in pancreas (~5%), bladder (~2.5%), skin (~2%), and brain (~1%) (Fig. 5F). Remarkably, however, the population of sulfated N-glycans in the pancreas was comprised exclusively of bi-antennary glycans, with and without core-fucose (e.g. $Hex_5HexNAc_4Fuc_1SO_4$). Other sulfated structures in the pancreas were found to also carry one or two additional alpha-Gal residues (e.g. $Hex_6HexNAc_4Fuc_1SO_4$), which aligned well with the high expression levels of alpha-Gal in the pancreas found by our study.

The expression of the fucosylated LacdiNAc N-glycan epitope was previously associated with different forms of cancer[65], and previous data has shown that LacdiNAc precursors can be fucosylated by FUT9[66], making it a candidate gene for regulating the tissue-specific biosynthesis of fucosylated LacdiNAc. Recently, fucosylated LacdiNAc containing structures have been identified in the N-glycome of the human brain[27] and of HEK-293 cell-line[67,68]. Systematic screening of all tissues for glycan precursors that generated the respective diagnostic fragment ion (HexNAc$_2$Fuc$_1$ – 553.2 amu) revealed that this epitope is expressed in a highly tissue-specific manner, with the highest levels found in kidney (~3%), colon (~2%), skin (~0.5%), and brain (~0.1%) (Fig. 5E). These observations were further corroborated by previous reports of *Fut9* expression in the proximal tubule of kidney, in intestinal epithelial cell and in neurons[44].

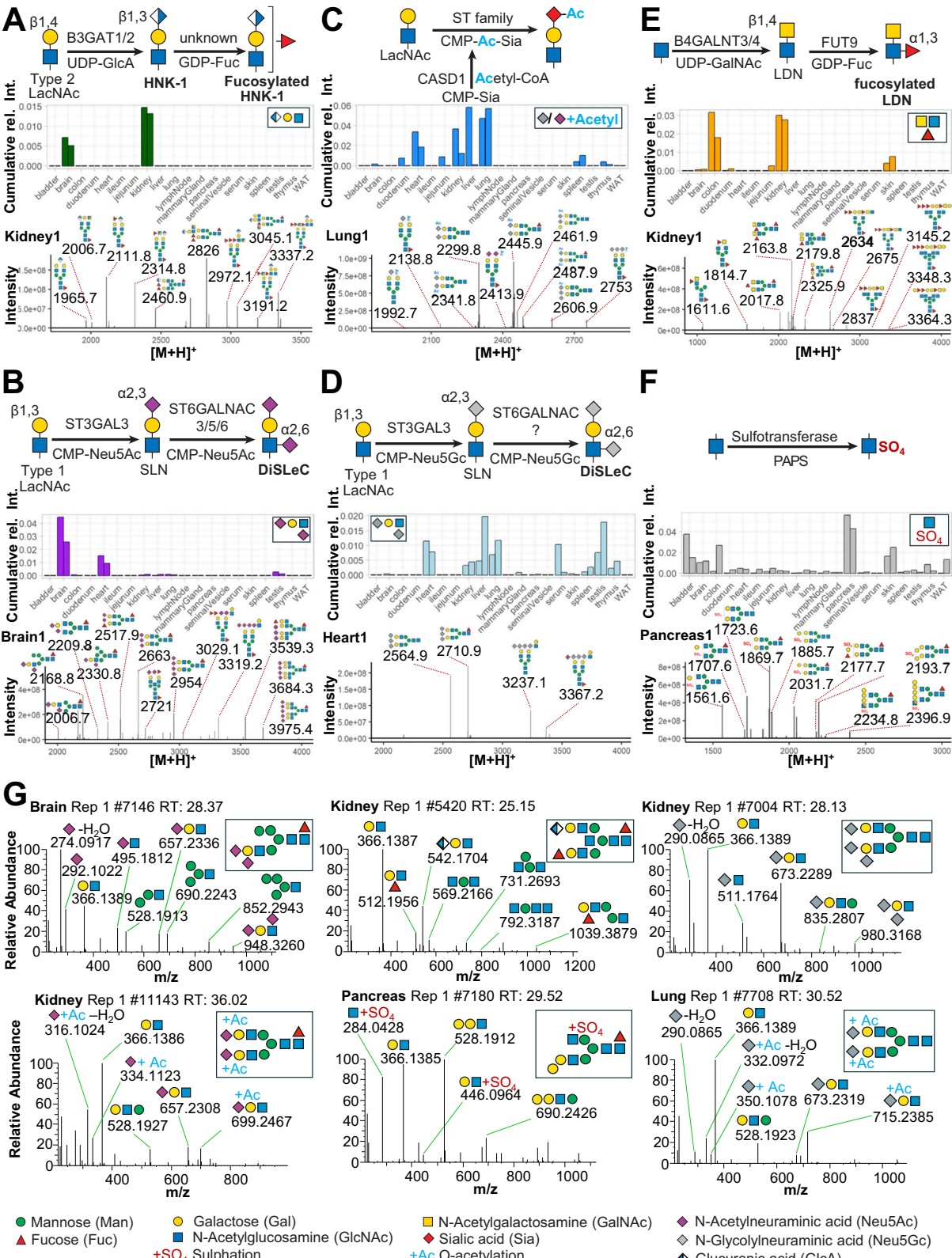

**Fig. 5 | Tissue specific expression of unusual N-glycans.** Comparative analysis of N-glycans carrying **A** HNK-1-epitope, **B** Neu5Ac-containing di-sialyl Lewis C-epitope, **C** O-Acetylated sialic acids, **D** Neu5Gc-containing di-sialyl Lewis C-epitope, **E** Fucosylated LacdiNAc and **F** Sulfated HexNAc. Representative N-glycan compositions bearing the respective modification are shown for selected tissues. N-glycan cartoons depict tentative structure assignments based on composition, using the Symbol Nomenclature for Glycans (SNFG). **G** Exemplary MS/MS data of representative N-glycan structures of the respective tissues. Diagnostic fragment ion compositions are indicated as cartoons, using the SNFG. WAT - white adipose tissue.

Neu5Ac and Neu5Gc can both be installed in either α2,3-, α2,6- or α2,8-linkage, to either galactose or -very rarely in N-glycans- to N-acetyl-hexosamines, such as N-acetyl-glucosamine (i.e. 6-sialyl Lewis C; as reported previously in small amounts in bovine fetuin[69]) or N-acetyl-galactosamine (e.g. sialyl LacDiNAc). We thus compared the relative abundances of the two sialic acid variants linked to hexoses, the canonical acceptor sites on N-glycans, or to N-acetylhexosamines (i.e. N-acetylglucosamine or N-acetylgalactosamine; 6-sialyl Lewis C, sialyl LacdiNAc, respectively), across all tissues. Again, we eSNOG-filtered all N-glycan derived MS/MS spectra for those that contained fragment ions diagnostic for sialic acids (i.e. fragment ion mass 292.1 amu and 308.1 amu) and for sialylated HexNAc (i.e. fragment ion mass 495.2 amu and 511.2 amu, respectively). As expected, the relative proportions of precursor signals of canonically sialylated N-glycans greatly exceeded those that were generated from the unusual sialyl-HexNAc structures, in all tissues. Also, the relative incorporation rates of Neu5Ac and Neu5Gc were essentially independent of the acceptor monosaccharide across all tissues. Remarkably, in brain Neu5Ac-HexNAc was found on approximately 10% of the sialylated N-glycan fraction.

We then systematically screened all tissues for fragment ions that are diagnostic for the doubly Neu5Ac-capped di-sialyl Lewis C epitope (i.e. 948.3 amu), which has previously been reported for brain[34,57], and its compositional analog, the Neu5Gc-decorated di-sialyl Lewis C epitope (i.e. 980.3 amu). Biosynthesis of di-sialyl Lewis C has been associated with the expression of the sialyltransferases ST6GALNAC3, ST6GALNAC5 and ST6GALNAC6, which are all able to catalyze the addition of α2,6 sialic acid to the GlcNAc residue within LacNAc motifs[70,71]. Interestingly, high expression level of ST6GALNAC5, which in healthy mouse and human is only found in the brain[72,73], has been strongly linked to brain-tropism of breast cancer metastasis[72]. This suggests that di-sialyl Lewis C may be involved in brain micro-environmental interactions facilitating metastatic niche establishment. Our cross-tissue analysis revealed that the expression of Neu5Ac-containing di-sialyl Lewis C is strictly limited to a small number of tissues, with the most prominent being brain (~3% of the TIC) and heart (~0.7%) (Fig. 5B). Only minor levels (<0.2%) of di-sialyl Lewis C N-glycans were found in kidney, testis, and lung. The associated precursor masses in the brain correspond to compositions ranging from potential hybrid-type (e.g. $Hex_5HexNAc_3Fuc_1Neu5Ac_2$, $Hex_6HexNAc_3Fuc_1Neu5Ac_2$) over biantennary complex-type N-glycans with up to four Neu5Ac-residues (e.g. $Hex_5HexNAc_4Fuc_1Neu5Ac_4$), to triantennary N-glycans with four Neu5Ac-residues (e.g. $Hex_6HexNAc_6Fuc_1Neu5Ac_4$). Neu5Gc-decorated variants of the di-sialyl Lewis C-epitope were detected, albeit at low levels, in multiple tissues. The highest expression levels were found in liver (ranging from ~2% to ~0.5%), heart (~1%), testis (ranging from ~2% to ~0.8%), lung (~0.8%), kidney (~0.4%), spleen (~0.2%), thymus (~0.4%), and serum (ranging from ~1% to ~0.1%) (Fig. 5D). Interestingly, the Neu5Gc-carrying di-sialyl Lewis C N-glycans exhibited a diminished structural complexity when compared to the Neu5Ac-capped variants found in the brain (Fig. 5). In contrast to their Neu5Ac-capped homologs, N-glycan compositions containing Neu5Gc-based di-sialyl Lewis C were almost identical across all relevant tissues, comprising essentially two biantennary N-glycans (i.e. $Hex_5HexNAc_4Neu5Gc_3$ and $Hex_5HexNAc_4Fuc_1Neu5Gc_3$).

Further dissecting the structural complexity of sialylated N-glycans, we also investigated the relative abundance of O-acetylated neuraminic acids across tissues. O-Acetylation of sialic acids is correlated with the circulatory half-life of glycoproteins in the human serum and can be crucial for their biological activities[74]. Most importantly, O-acetylated sialic acids are critical entry receptors for many respiratory viruses, including Influenza C virus, human coronavirus OC43 and the murine coronavirus[75]. Murine coronaviruses often spread to the liver, an organ topism that has been suggested to be partially explained by the expression pattern of Ac-Neu5Ac and Ac-Neu5Gc in both lung

and liver[76]. Quantifying the signals of all N-glycan precursors with O-acetylated neuraminic acids (i.e. Ac-Neu5Ac, fragment ion mass = 334.1 amu and/or Ac-Neu5Gc, fragment ion mass = 350.1 amu) allowed us to confirm expression of these important modifications almost exclusively in five tissues, namely lung (~4.5%), heart (~2%), kidney (~1.5%), and, at very low levels, spleen (<0.1%) (Fig. 5C). Liver gave ambiguous results, as liver 1 showed very high, and liver 2 lower expression of acetylated sialic acids (~3.5% and ~0.2%, respectively). The respective compositions in the lung are all partially core-fucosylated, bi-antennary N-glycans with one, two, or three Neu5Gc- or Neu5Ac-residues, of which one sialic acid residue was acetylated (e.g. $Hex_5HexNAc_4Fuc_1Neu5Gc_1Ac_1$). So far, CASD1 is the only mammalian enzyme known to catalyze the acetylation of sialic acid resulting in the formation of Neu5,9Ac2[77]. In mouse, like human, CASD1 appears to be widely expressed among most organs[44]. Our findings suggest additional unknown regulatory mechanisms that restrict the expression of O-acetylated neuraminic acids to specific organs. Furthermore, we predominantly observed O-acetylation on Neu5Gc and only to a low degree on Neu5Ac residues (Supplementary Fig. 7H and Supplementary Fig. 7I). This suggests important differences in the biosynthesis, stability, or incorporation of the two O-acetylated sialic acid variants into N-glycans. As it is unclear whether CASD1 also catalyzes the O-acetylation of Neu5Gc, this warrants further investigations into the substrate specificities of CASD1 and the identification of additional O-acetyltransferases[78].

## Profiling the isomeric structural complexity of the murine N-glycome

Next to compositional variations, N-glycans exhibit a unique level of micro-heterogeneity that derives from structural, positional, and anomeric isomers. The number of unique N-glycan structures that can be constructed from a given mono-saccharide composition (hence of identical molecular mass, i.e. isobaric) adds a critical layer of complexity to the N-glycome[9]. Importantly, the co-existence of multiple isobaric N-glycan isomers within a single tissue cannot be captured by MS (or even MS/MS) alone and can only be resolved by either chromatographic or ion-mobility-based separation techniques. To generate isomer-sensitive information, in this study, all samples were analyzed using a highly isomer-selective stationary phase (PGC-LC). Previously established N-glycan retention libraries, knowledge of elution properties of N-glycans on PGC and MS/MS data were used to identify the exact structures of respective isomers[9,26,27,79].

Expanding our N-glycome analyses by the integration of chromatographic information revealed the staggering structural complexity of the mouse N-glycome and provided insight into vital, organ-intrinsic regulation of glycobiological pathways. To showcase the granularity of our N-glycan isomer-sensitive dataset, we first compared the retention times of a specific, single precursor mass, that holds all doubly Neu5Ac-sialylated, core-fucosylated, biantennary, complex type N-glycan structures of composition $Hex_5HexNAc_4Fuc1Neu5Ac_2$, across tissues (Fig. 6A). As sialic acids are usually found in either α2,3- or α2,6-linkage to terminal galactose residues of N-glycans, up to four different structural isomers of unique retention times are expected for this single glycan composition: both antennae carrying α2,3-linked, both antennae capped with α2,6-linked, and one of the antennae with α2,6-linked while the other antenna bears a α2,3-linked Neu5Ac. To compensate for experimental chromatographic elution-time shifts between samples, all retention times within a given analytical run were normalized to those of the consistently detected Man5 N-glycan structure. Based on previously established, differential retention properties of sialylated N-glycans on PGC-chromatography, for which α2,3-linked Neu5Ac exhibit stronger retention than α2,6-linked Neu5Ac, we were able to identify the underlying Neu5Ac-linkages[79,80].

Elution profiles of N-glycan structures that comprised of $Hex_5HexNAc_4Fuc_1Neu5Ac_2$ showed that almost all tissues were

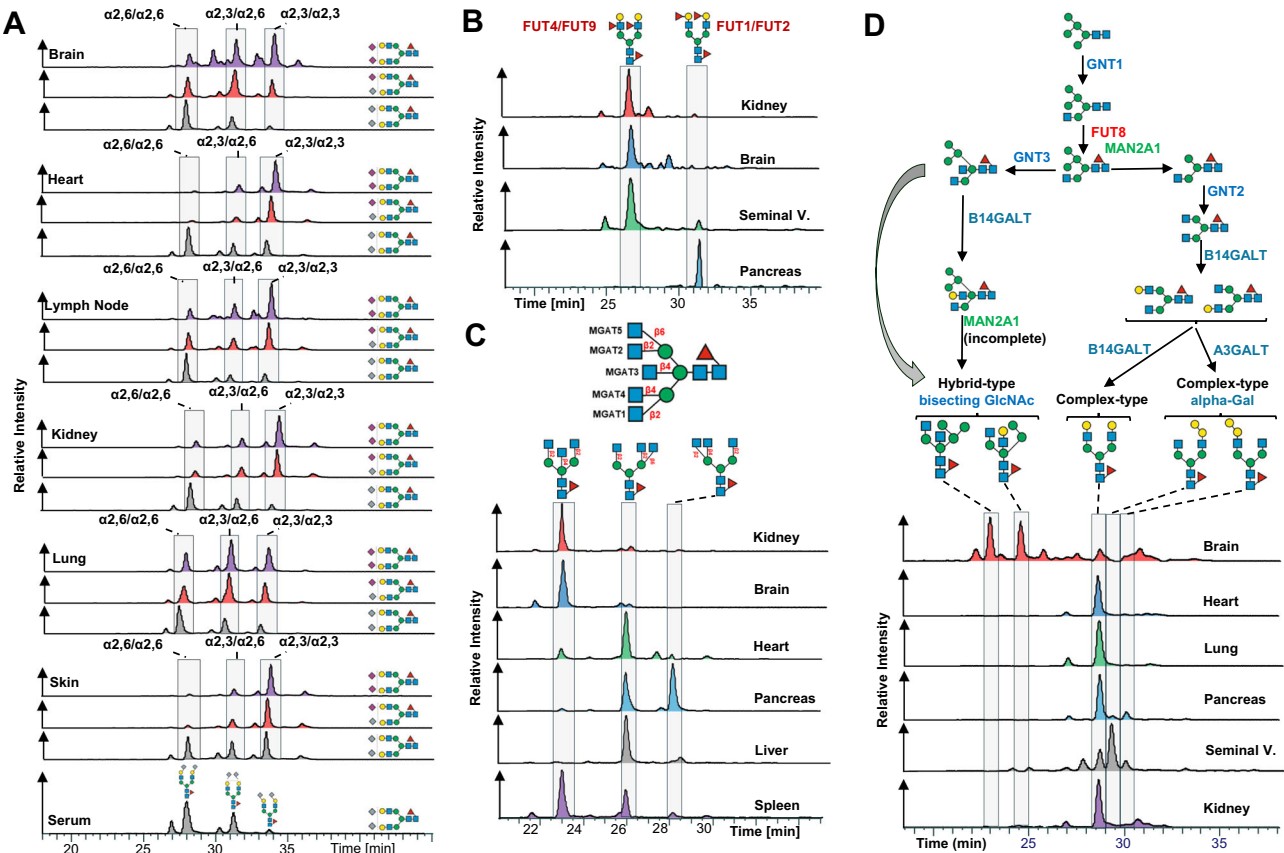

**Fig. 6 | Isomer-specific profiling of the murine N-glycome.** Closely related structural N-glycan isomers were separated with PGC-LC. N-glycan retention libraries in conjunctions with MS/MS data were used to identify the exact structure of the respective isomers. All retention times were normalized to the Man5 N-glycan. **A** Elution profiles for the compositions Hex₅HexNAc₄Fuc₁Neu5Ac₂ (purple), Hex₅HexNAc₄Fuc₁Neu5Ac₁Neu5Gc₁ (red), and ex₅HexNAc₄Fuc₁Neu5Gc₂ (light blue) across selected tissues. **B** Elution profiles for the composition

Hex₃HexNAc₅Fuc₁ across kidney (red), brain (blue), heart (green), pancreas (orange), liver (grey), and spleen (purple). **C** Elution profiles of the heavily fucosylated composition Hex₅HexNAc₄Fuc₃ in the kidney (red), brain (blue), seminal vesicle (green), and pancreas (orange). **D** Biosynthetic pathway leading to complex-type and hybrid-type N-glycans and elution profiles of the composition Hex₅HexNAc₄Fuc₁ in the brain (red), heart (blue), lung (green), pancreas (orange), seminal vesicles (grey), and kidney (purple).

dominated by α2,3-linked Neu5Ac (Fig. 6A, Supplementary Fig. 9D). Interestingly, however, brain, lung, and testis presented with a balanced ratio of α2,3- and α2,6-linked Neu5Ac isomers. Moreover, the brain displayed a notable presence of distinct N-glycan structure isomers due to the occurrence of branching Neu5Ac and/or antennary fucose. Remarkably, these specific glycan structures were exclusive to the brain tissue and were not identified in any other organ within the mouse.

Next, we compared the elution profile of the corresponding Neu5Gc-sialylated N-glycan structures (i.e. Hex₅HexNAc₄Fuc₁Neu5Gc₂, Supplementary Fig. 9E) across all tissues. Of note, the combined masses of Neu5Gc and Fuc are identical to the combined masses of Neu5Ac and Hex, precluding the discrimination of glycans containing Neu5Gc and Neu5Ac by MS alone. Importantly, however, on the same glycan backbone, Neu5Gc-residues lead to a slight retention time shift compared to Neu5Ac-residues on PGC[80], which thus allowed us to unambiguously assign Neu5Gc-containing glycans and their respective linkage. In stark contrast to Neu5Ac-sialylated structures, Neu5Gc-sialylated structures were predominantly found in α2,6-linkage across tissues, except for skin (i.e. ratio of α2,3- to α2,6-linked Neu5Gc approx 50%) (Fig. 6A, Supplementary Fig. 9E). This entirely different construction of the Neu5Gc N-sialome compared to its Neu5Ac-terminated counterpart raised the question of the origin of these differently sialylated structures.

The elution profile of the mixed Neu5Ac/Neu5Gc composition (Hex₅HexNAc₄Fuc₁Neu5Ac₁Neu5Gc₁), closely resembled the elution

profile of the Neu5Ac/Neu5Ac-sialylated structures in all tissues (Fig. 6A), suggesting a shared origin for these structures. By contrast, the structural profile of entirely Neu5Gc-sialylated structures (i.e. Hex₅HexNAc₄Fuc₁Neu5Gc₂) markedly deviated from the Neu5Ac/Neu5Gc and Neu5Ac/Neu5Ac patterns. Notably, the elution profile of Neu5Gc/Neu5Gc structures exhibited minimal variation across tissues and closely resembled the serum elution profile, suggesting that these structures were actually derived from (contaminant) serum glycoproteins.

PGC-LC also allowed us to discriminate pivotal positional differences in distal fucosylation (i.e. α1,2-linked to galactose or α1,3-linked to GlcNAc-residues) that give rise to the important, glycan-associated immune-determinants bgH and Lewis X. While Lewis X determinants are mainly synthesized by FUT4 or FUT9, bgH-epitopes are synthesized by FUT1 or FUT2[81]. To discern these critical fucose structures, we compared the retention time profiles of a multiply fucosylated precursor mass of the compositions Hex₅HexNAc₄Fuc₃ across all relevant samples (Fig. 6B). The N-glycan structures that may be deduced from this single composition comprise asialo, core-fucosylated, biantennary, complex type N-structures with two distal fucoses, either in bgH- or Lewis X-related linkage. Based on our normalized elution profile data we found unexpected differences in antennary fucose between brain, kidney, seminal vesicles, and pancreas (Fig. 6B). Previous studies showed that the attachment of α1,3-linked fucose residues leads to pronounced shorter retention times on PGC, while the attachment of α1,2-linked fucose residues or the attachment of α1,6-core-fucose

leads to stronger retention[9,26,27]. While brain and kidney were found to contain almost exclusively the Lewis X-containing isomer[82], the pancreas exhibited only the bgH-epitope-containing isomer, which correlates with the high expression of FUT1 in the pancreas[83]. A seminal vesicle, however, exhibited signals corresponding to both the Lewis X and the bgH-eptitope containing isomer. This finding aligns well with the occurrence of the Lewis Y-epitope, as described above, and the high expression level of FUT1, FUT2 and FUT4 in seminal vesicle[42]. Interestingly, a major carrier of Lewis X and Lewis Y in human seminal plasma has been linked to Glycodelin isoform S (GdS)[84]. As Lewis X and Y are known ligands to the immune receptor DC-SIGN[85] and glycodelins are potent immunosuppressors, they have been suggested to be important for feto-embryonic defense[86] against adverse immune reactions in the early stages of gestation.

Chromatography also provided stratification of arm/branch-isomers of tri-antennary, complex-type structures. Complex-type N-glycans are characterized by the substitution of both (i.e. α1,3- and α1,6-) terminal core-mannoses by the addition of GlcNAc residues by *Mgat1* and *Mgat2*, respectively. Additional branching of such complex-type structures results from secondary GlcNAc substitutions on the (i.e. α1,3- and α1,6-) terminal core-mannoses by *Mgat4* and *Mgat5*, respectively, or the installation of a bisecting GlcNAc to the core-central mannose by *Mgat3*. Our isomer-sensitive analysis thus allowed us to discern between the catalytic activities of the five different glycosyltransferases involved across all tissues. To explore the occurrence of biologically distinct N-glycan structures of identical composition, we compared the retention times of the precursor composition $Hex_3HexNAc_5Fuc_1$ across all tissues which could either be multiple tri-antennary structural isomers or a single biantennary, bisected complex type structure (Fig. 6C, Supplementary Fig. 9C). The respective elution profiles indicated three dominant peaks. As established previously, the attachment of a bisecting GlcNAc-residue leads to a pronounced shortening of the retention time on PGC, while the addition of a branching GlcNAc increases the retention time[26,27].

Intriguingly, brain, kidney, colon, ileum, duodenum, jejunum, and serum were strongly dominated by biantennary, bisected structure isomers. While this confirmed the results of our MS/MS-based profiling, it also suggests a high occurrence of bisecting GlcNAc in the brain, kidney, and colon, corroborated by *Mgat3* expression data[87].

Apart from oligomannose-type, N-glycans are often classified into hybrid-type or complex-type structures. Hybrid-type N-glycans result from the incomplete action of alpha-mannosidase II, giving rise to unsubstituted mannose residues on the α1,6-arm and a potentially substituted GlcNAc residue linked to the α1,3-mannose residue[1]. Interestingly, many N-glycome studies infer N-glycan classification based on composition. We found that this approach is highly oversimplifying the experimental data presented here. For example, the composition $Hex_5HexNAc_3Fuc_1$, which is often considered an archetypical hybrid type N-glycan structure, separated into several chromatographic peaks across tissues (Supplementary Fig. 9A). Although most tissues exhibited highly similar elution profiles, pancreas, seminal vesicle, thymus, and spleen showed distinct dominant structures, which eluted much later, indicative of the presence of alpha-galactose (as the addition of alpha-galactose increases the retention time[9,88]). Manual inspection of the associated MS/MS spectra indicated core-fucosylated, truncated N-glycan structure with an alpha-galactosylated antenna (pronounced 528 m/z fragment ion[9,88]). This was further corroborated by high levels of alpha-galactose in the pancreas, seminal vesicle, thymus, and spleen, as were observed in our initial MS/MS based N-glycome profiling, and implied elevated levels of hexosaminidase in these tissues.

The classification of N-glycan structures, solely relying on composition, is further complicated by the incorporation of bisecting GlcNAc into hybrid-type N-glycans. Many hybrid-type structures with bisecting GlcNAc have counterparts in the class of complex-type N-glycans, adding an additional layer of complexity to their delineation based on composition[9] (e.g. $Hex_5HexNAc_4Fuc_1$, $Hex_6HexNAc_4Fuc_2$, $Hex_5HexNAc_4$, $Hex_6HexNAc_4$). While the classification of the composition $Hex_5HexNAc_4Fuc_1$ as complex-type was true for most tissues we analyzed, we found that brain exhibited at least two hybrid-type structures within this composition, which we have also identified in human brain[9] (Fig. 6D, Supplementary Fig. 9A). Similarly, while the early retention time and the underlying MS/MS spectra showed that the composition $Hex_6HexNAc_4Fuc_1$ consisted of two hybrid-type structures containing bisecting GlcNAc in the brain, it represented alpha-galactosylated complex-type structures when present in other tissues (pronounced 528 m/z fragment ion, late elution time) (Supplementary Fig. 10). The same was found for the composition $Hex_6HexNAc_4Fuc_2$, which consists of hybrid-type structures with a bisecting GlcNAc in the brain, yet complex-type structures when present in other tissues (Supplementary Fig. 10B). These findings suggest that hybrid-type structures with bisecting GlcNAc are yet another highly distinctive feature of the brain N-glycome, and they boldly underscore the merits of isomer-sensitive N-glycome analyses to uncover unexpected structural signatures.

## Discussion

Here we make publicly available a consistent and structure-sensitive N-glycome dataset of 20 different mouse tissues, essential for modern integrative system biology efforts. We used PGC-LC to chromatographically separate even closely related N-glycan isomers, an Orbitrap mass-analyzer for high-resolution tandem mass-spectrometric data-acquisition, and an unconventional, non-targeted data analysis approach. Our entirely data-driven, analytical approaches provide means to automatically extract and query semiquantitative glycan profiles. It allows for the rapid and comprehensive N-glycomic analysis of large sample sets, such as the here presented mouse atlas. We complemented this dataset with established methods of isomer analysis through PGC retention time, adding isomer-specific tissue signatures, which provide additional layers of glycan information.

Our workflow provides an unprecedented level of depth and structural fidelity, allows for the identification of multiple times more glycan features than previous studies, and highlights organ-intrinsic regulations of glycobiological pathways of not yet fully understood functionality. Additionally, using our approach, we identified and consistently compared the expression of rare, non-canonical modifications, even enabling the discovery of previously unreported modifications.

Unsupervised data analysis found clustering and convergent N-glycan signatures even among distantly related mouse tissues. These clustering patterns provide impartial insights into the glycosylation variations within organ groups, which potentially reflect their functional relationships or physiological roles.

Furthermore, our data suggests that different tissues use different biocatalytic strategies to specify their glycobiological identity, and thereby generate the staggering complexity revealed by our study. For example, while brain N-glycan diversity mainly results from its unique capacity to generate highly unusual and tissue-specific monosaccharide linkages to complex- and hybrid-type N-glycan core structures (e.g. branching Neu5Ac or HNK-1; Fig. 5), the structural complexity of lung N-glycans largely draws from the incorporation and modification of different sialic acid variants (e.g. differently O-acetylated Neu5Gc and Neu5Ac). By contrast, exocrine organs, such as the pancreas or seminal vesicles, shape their diversity of N-glycan structures primarily via a uniquely tuned interplay of alpha-galactosylation and antennary fucosylation. Although large parts of such tissue-specific N-glycome signatures can be aligned with gene expression data, the expression of non-canonical N-glycan structures implies the existence of additional, hitherto unknown, regulatory

mechanisms that restrict the catalytic activity of glyco-enzymes to specific organs.

Taken together, using a non-targeted isomer-sensitive workflow to characterize a consistent and essentially complete mouse N-glycome atlas, our analyses reveal tissue-specific N-glycan-structural features and highlight important organ-intrinsic regulations of glycobiological pathways. We anticipate that both our dataset and our analytical workflow will be instructive for fundamental glycobiological research, glycan analytical benchmarking, the development of glycome data analysis tools, and integrative systems glycobiology.

## Methods

### Preparation of mouse tissues
All mice were bred, maintained, examined, and euthanized following the recommendation of the Institutional Animal Welfare Committee and in accordance with the Austrian and European legislation. Two male and two female C57BL/6 J mice were obtained from the Jackson Laboratory (PRID: IMSR_JAX:000664, Bar Harbor, ME). C57BL/6 J mice were bred in the licensed breeding facility of the Institute of Molecular Biotechnology (IMBA, Vienna, AT), under a 14hrs/10hrs light/dark cycle. Food and water were available ad libitum. All mice were euthanized with carbon dioxide at the age of 13 weeks. Collected tissues are detailed in Supplementary Table 3. Prior to freezing, duodenum, jejunum, ileum, colon, and bladder were opened and cleaned with PBS for food, fecal matter and/or urine. Serum was collected from blood with Microtainer SST tubes (BD Biosciences, 365968). All samples were immediately snap-frozen in liquid nitrogen after collection and stored at −80 °C until further processing.

### N-glycan extraction
Each mouse tissue was transferred into a Falcon tube and mixed with 100 mM ammonium bicarbonate buffer containing 20 mM ditiothreitol and 2% sodium dodecyl sulfate in a total volume of 2 mL. After homogenization with an Ultra-Turrax T25 disperser, the homogenate was incubated at 56 °C for 30 min. After cooling down, the solution was brought to 40 mM iodoacetamide and incubated at room temperature in the dark for 30 min. After centrifugal clarification at 4000 × $g$, chloroform-methanol extraction of the proteins was carried out using standard protocols[66]. For this, the supernatant was mixed with four volumes of methanol, one volume of chloroform, and three volumes of water in the given order. After centrifugation of the mixture for 3 min at 4000 × $g$, the upper phase was removed, 4 mL of methanol were added, and the pellet was resuspended. The solution was centrifuged at 4000×g, the supernatant was removed, and the pellet was again resuspended in 4 mL methanol. The last step was repeated two more times. After the last methanol washing step, the pellet was dried at room temperature. The dried pellet was taken up in 50 mM ammonium acetate o (pH 8.4). For N-glycan release 2.5 U N-glycosidase F was added and the resulting mixture was incubated overnight at 37 °C. The reaction mixture was acidified with drops of glacial acetic acid and centrifuged at 4000 × $g$. The supernatant was loaded onto a Hypersep C18 cartridge (1000 mg; Thermo Scientific, Vienna) that had been previously primed with 2 mL of methanol and equilibrated with 10 mL water. The sample was applied, and the column was washed with 4 mL water. The flow-through and the wash solution were collected and subjected to centrifugal evaporation. Reduction of the glycans was carried out in 1% sodium borohydride in 50 mM NaOH at room temperature overnight. The reaction was quenched by the addition of two drops glacial acetic acid. Desalting was performed using HyperSep Hypercarb solid-phase extraction cartridges (25 mg) (Thermo Scientific, Vienna). The cartridges were primed with 450 μL methanol followed by 450 μL 80% acetonitrile in 10 mM ammonium bicarbonate. Equilibration was carried out by the addition of three times 450 μL water. The sample was applied and washed three times with 450 μL water. N-glycans were eluted by the addition of two times 450 μL 80% acetonitrile in 10 mM ammonium

bicarbonate. The eluate was subjected to centrifugal evaporation and the dried N-glycans were taken up in 20 μL HQ-water. 5 μL of each sample were subjected to LC-MS/MS analysis.

### LC-MS/MS analysis
LC-MS analysis was performed on a Dionex Ultimate 3000 UHPLC system coupled to an Orbitrap Exploris 480 Mass Spectrometer (Thermo Scientific). The purified glycans were loaded on a Hypercarb column (100 mm × 0.32 mm, 5 μm particle size, Thermo Scientific, Waltham, MA, USA) with 10 mM ammonium bicarbonate as the aqueous solvent A and 80% acetonitrile in 50 mM ammonium bicarbonate as solvent B. The gradient was as follows: 0–4.5 min 1% B, from 4.5 – 5.5 min 1–9% B, from 5.5–30 min 9–20% B, from 30–41.5 min 20–35% B, from 41.5–45 min 35–65% B, followed by an equilibration period at 1% B from 45–55 min. The flowrate was 6 μL/min. MS analysis was performed in data-dependent acquisition (DDA) mode with positive polarity from 500–1500 m/z using the following parameters: resolution was set to 60,000 with a normalized AGC target of 300%. The 10 most abundant precursors (charge states 2-6) within an isolation window of 1.4 m/z were selected for fragmentation. Dynamic exclusion was set at 20 s ($n = 1$) with a mass tolerance of ± 10 ppm. Normalized collision energy (NCE) for HCD was set to 20, 25, and 30% and the precursor intensity threshold was set to 8000. MS/MS spectra were recorded with a resolution of 15,000, using a normalized AGC target of 100% and a maximum accumulation time of 100 ms.

### Data analysis
For the MS/MS based profiling, MS2 data (in the.raw file format) were extracted, refined (i.e. precursor mass and charge-state; no scan merging), and converted into.mgf file format using PEAKS[49] X Pro Studio 10.6 (build 20201221) or converted into.mgf files using the vendor-specific peak-picking algorithm implemented in MSConvert (version 3.0.22133-5eed1a6) and removing zero-samples. MS2 spectra were filtered for (N-)glycan specific oxonium-ions (i.e. 167.0914, 183.0863, 204.0867, 222.0972, 224.1118, 243.0264, 274.0921, 284.0435, 290.087, 292.1027, 308.0976, 312.1289, 316.1027, 328.1238, 332.0976, 334.1133, 350.1082, 366.1395, 370.1697, 407.1661, 495.1821, 511.177, 512.1974, 528.1923, 542.1716, 553.224, 569.21887, 622.1284, 657.2349, 658.2553, 673.2298, 698.2615, 699.2455, 714.2564, 715.2404, 715.27677, 792.3234, 803.2928, 819.2877, 860.31427, 876.30917, 948.3303, 980.3201, 446.09627, 649.17567, 731.27167, 877.32957, 1023.38747, 1169.44537; mass-tolerance +/- 0.05 amu) using a custom-made perl-script (Supplementary Software "MS2OxoPlot.pl"), binned according to retention time (bin size = 10 s), counted and visualized using R.

MS1 data were extracted, charge-deconvoluted, and deisotoped using Decon2[50] with custom parameters adjusted for N-glycomics data (Supplementary Data 1). In the next step, we assigned the precursor mass and intensity values to mass bins (i.e. in the range of 1000 - 5000 Da; bin-width = ±0.05 amu), summed the individual precursor mass-to-intensity values across the entire chromatographic time-range (i.e. retention time 0 – 50 min) and eventually removed signals below a cumulative intensity threshold of 5E + 6.

Raw MS/MS data were refined (i.e. mono-isotopic peak picking and charge state assignment) and converted into the generic.mgf file-format using PEAKS[49]. Analogous to MS1-data processing, the refined precursor-information was deconvoluted and assigned into mass-bins (bin-width = ±0.05 amu). From this, to automatically identify precursors that derived from N-glycans and to stringently control for unintended precursor ion co-isolation events, we calculated mass-bin-specific score-values (SNOG-score) from the intensity value of an N- (and O-) glycan-specific fragment ion (i.e. oxonium ion of the reduced-end monosaccharide GlcNAc; [M + H]+ = 224.1 amu) using custom-code. To remove signals derived from non-N-glycan-associated structures, all mass-bins with a SNOG-score lower than 0.03 were rejected. This step was performed sample-specifically. The resulting inclusion

list was used to retrieve the precursor-intensity information from the processed MS1 data, using R.

To stratify the dataset based on sub-structural modifications (e.g. sialylation, fucosylation, sulfation), we used diagnostic fragment ions known from previous studies[27,55,89] and screened the dataset for so-far unknown reporter-ions (Supplementary Table 2). The sub-structure-specific extended SNOG-score (eSNOG) is calculated analogous to the SNOG-score, but instead of using the N-glycan specific fragment-ion 224.1, sub-structure-specific fragment ions are used (Supplementary Table 2). As the different diagnostic fragment ions exhibit vastly differing relative intensities within a MS/MS-spectrum, the cut-off values for the eSNOGs are empirically determined for each of the fragment ions. The resulting sample- and sub-structural feature-specific inclusion list was used to retrieve the precursor-intensity information from the processed MS1 data, using R.

All data visualization and statistical analyses were performed with R 4.3.1 using the packages "tidyverse"[90] (2.0.0), "pheatmap" (1.0.12), "RColorBrewer" (1.1.3), "dendextend "(1.17.1)[91], "corrr" (0.4.4), "ggrepel" (0.9.3), and "Rtsne" (0.17).

### Reporting summary

Further information on research design is available in the Nature Portfolio Reporting Summary linked to this article.

## Data availability

The raw mouse N-glycome PGC-LC-MS/MS data (.raw) and processed datasets referred to in this manuscript (in the.csv file format) are publicly available via the GlycoPOST repository[92] (Dataset ID: GPST000398; https://glycopost.glycosmos.org/entry/GPST000398) and the GitHub repository (https://github.com/holandir/Mouse-N-glycome-analysis). Tissue and sample replicate numbers are indicated in the respective file names. Source data for the graphs presented in the main article are provided with this paper as a Source Data file. Source data are provided with this paper.

## Code availability

All custom code, R scripts, and software for MS and MS/MS data pre-processing is freely available at: http://homepage.boku.ac.at/jstadlmann and https://github.com/holandir/Mouse-N-glycome-analysis.

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

## Acknowledgements

J.H. was partly funded by the European Commission (Newcotiana 760331). S.M. is funded by the ESPRIT-Program of theAustrian Science Fund (FWF, Project number: ESP 166). D.M.M. is supported by a grant from the National Institutes of Health, R01AI175124. J.M.P. received funding from the Medical University of Vienna, the T. von Zastrow foundation, the Canada 150 Research Chairs Program F18-01336 and the Innovative Medicines Initiative 2 Joint Undertaking under grant agreement No 101005026. This Joint Undertaking receives support from the European Union's Horizon 2020 research and innovation program and EFPIA. We also gratefully acknowledge funding by the German Federal Ministry of Education and Research (BMBF) under the project "Microbial Stargazing - Erforschung von Resilienzmechanismen von Mikroben und Menschen" (Ref. 01KX2324). This project was supported by MS equipment kindly provided the EQ-BOKU VIBT GmbH and the BOKU Core Facility Mass Spectrometry.

## Author contributions

J.H., S.M. and J.S. conceived the project and wrote the manuscript. J.H. and SM prepared samples, analyzed data, and assisted in the interpretation of the results. J.H. performed mass-spectrometry analyses and assisted in algorithm development. J.S. wrote software and analyzed data. T.O. and F.A. analyzed data and assisted in the interpretation of the results. A.G. assisted with initial algorithm development and software design. D.M., J.P. F.A. and J.S. discussed the study design, and results, drafted the manuscript and supervised the project.

## Competing interests

The authors declare no competing interests.
