## [Transparent Peer Review file · Nature Communications]

Non-targeted N-glycome profiling reveals new layers of organ-specific diversity in mice

Corresponding Author: Dr Johannes Stadlmann

Version 0:

Reviewer comments:

Reviewer #1

(Remarks to the Author)

This manuscript by Helm et al. presents a large N-glycome dataset comprising 20 different mouse tissues, generated through PGC-LC-MS/MS analysis and described a data analysis workflow for interpreting these data. The unbiased determination of glycans across different organisms has long been a necessity and a challenge. The substantial corpus of publicly available N-glycome data contributed by this study is valuable to the fields of glycomics and glycobiology. However, the impact of this work has been diminished by the lack of rigorous and transparent data interpretation, thereby undermining the credibility of the conclusions drawn from these results.

The primary concern is that the title of the manuscript suggests it presents an analysis method capable of distinguishing between glycan isomers. However, actually, only about 1/6 of the content discusses approach for discriminating glycan terminal monosaccharides, such as sialic acid, based on chromatographic retention times. This approach is not particularly novel and is commonly found in other glycomics analyses. Moreover, the glycome analysis in this work involves direct extraction of unstructured information from mass spectrometry data without hierarchical processing or quality control. Consequently, it cannot serve as a comprehensive framework for glycan composition/structure-level mass spectrometry data analysis. Each step of data interpretation is conducted independently, lacking a systematic process for data retrieval and analysis. For instance, in the first section, "Precursor-independent MS/MS-based N-glycome profiling of 20 mouse tissues," the authors primarily extract information based on specific fragment ions, inferring differences among tissues based on ion intensities. However, such quantification is rudimentary and overlooks factors like fragmentation efficiency in mass spectrometry. The filtering method used in this work based on SNOG-scoring and eSNOG-scoring relies on empirical score. It's worrisome whether this filtering method is universally applicable to all glycome data, given the significant variations in glycomics analysis.

Other concerns include:

- 1) The authors stated that each sample was analyzed in duplicates, but when conducting statistical analysis of different glycan profiles among tissues, such as in Fig.1, SFig.1, and SFig.2, the differences and significance were not statistically analyzed.
- 2) The authors employed the diagnostic ion for reduced N-acetylhexosamine to filter spectra, obtaining 220,506 spectra from 509,283 spectra. Was the intensity threshold of 224m/z considered during filtration? Additionally, were the filtered-out spectra containing other glycan fragments? The authors utilized glycan diagnostic ions to showcase the abundance of a particular glycan type within a sample. However, the sources of glycan fragments for a specific type are not singular. Taking Neu5Ac-containing glycans as an example, while it's accurate to consider 292 Da and 274 Da, there could also be fragments like 495 Da. How did the authors calculate the proportion of Neu5Ac-containing glycans in a sample?
- 3) Utilizing PGC retention times for isomer analysis isn't a novel method. What specific optimizations did the authors make regarding PGC? What advantages does your approach offer for isomer analysis compared to existing methods? Regrettably, in this manuscript, apart from applicability to larger datasets, I didn't see any other advancements.
- 4) The authors mentioned that their analysis can address issues such as incorrect mono-isotopic peak-picking or charge-state assignments by the mass-spectrometer, unintended precursor ion co-isolation, in-source fragmentation, or excessive adduct-ion formation (Line 189-191). However, they didn't provide corresponding evidence through benchmark data or comparisons to demonstrate how these issues were addressed or the extent of improvement achieved.
- 5) In this study, the authors calculated that almost 50% of the brain N-glycome is sialylated. It's noteworthy that in other studies, proportions of 3%, 20%, and 40% have been reported (line 323). What factors could lead to such significant differences? Was it due to previous methods lacking sensitivity, experiencing in-source fragmentation, or difficulty in detecting sialic acid? The author not only failed to present a detailed comparison of methods but also didn't engage in

thorough discussion, attributing the differences solely to methodological variations, which is insufficiently rigorous. How should one choose a method they want to research the content of specific glycosylation?

6) There are minor changes needed as well, such as replacing "amu" with "Da," which is more commonly used in current mass spectrometry analysis, or "m/z." However, this change is not imperative.

Reviewer #2

(Remarks to the Author)

The paper aims to characterize the diversity and heterogeneity of N-glycome across various tissues in mouse. Instead of the profiling of specific N-glycans, the paper developed computational methods for profiling classes of N-glycans (e.g., sialylated, sulfated and fucosylated glycans, or the glycans with different linkage of sialic acids). The profiling methods are based on the MS/MS spectra of glycans containing signature oxonium-ions for different classes, which is easy to be implemented and applied. The paper also presented the relative abundance profiles of isomeric peaks based on the retention profiles under LC. The analyses across 20 mouse tissues revealed specific glycan patterns in different tissues (e.g., in brain vs. other tissues).

The computational methods used in the study is simple but interesting. Using signature oxonium-ions to classify MS/MS spectra of N-glycans have been long used, in particular to filter glycan spectra in high-throughput analyses. This seems to a comprehensive study based on this approach leading to interesting findings. However, the accuracy of the method has not been thoroughly validated previously. So it is useful for the authors to first establish a baseline of potential errors of such approach. In particular, the intensities of oxonium-ions appears to be highly correlated with the experimental condition such as fragmentation energy. It is not uncommon that some MS/MS spectra of glycans miss certain oxonium-ions. It is unclear if the classification based solely on signature oxonium-ions (i.e., the SNOG-score) will result in inaccurate estimation of different classes of N-glycans in the complex sample. It would be useful to test this method using standard samples to estimate the level of variance across samples and across N-glycan classes due to inaccurate estimation.

Some methods were not completely clear from the methodology description. For examples, how are the glycan abundance profiles in Fig 1 estimated? Are they based on the total intensity of the precursor ions (in MS1) classified in each class of N-glycans? If that is case, does it assume different classes of N-glycans (e.g., sialylated vs. others) have similar responses in MS? Also, How are the isomeric glycans with different linkages identified in the retention profile (e.g., in Fig. 6).

Finally, it is unclear if and how the analysis were validated through biological/technical replicates. In In 233-234, it is stated that "... validated robust reproducibility among sample replicates ...", but no details were given.

Reviewer #3

(Remarks to the Author)

The presented manuscript by Helm et al. provides a consistent N-glycome dataset of 20 different mouse tissues and demonstrates a multimodal data analysis workflow that allows for outstanding depth and coverage of N-glycome features. This highly scalable, LC-MS/MS data-driven method integrates the automated identification of N-glycan spectra, the application of non-targeted N-glycome profiling strategies, and the isomer-sensitive analysis of glycan structures. In addition, the manuscript also provided N-glycome features for critical sub-structural determinants and glycan isomers across the mouse N-glycome, uncovering tissue-specific glycosylation patterns and highlighting multiple layers of N-glycome complexity that derive from organ-specific regulations of glycobiological pathways. Overall, the manuscript is well-written and has compelling data. This is a vast data mining effort consolidating 20 different mouse tissue N-glycome profiling and provides automated identification and annotation of N-glycan spectra as opposed to conventional PGC-LC-MS/MS manual analysis that relies on the targeted, selective extraction for known or anticipated glycan compositions and masses, and subsequent structure assignment using pre-established, relative chromatographic retention times, complementary MS/MS data analysis, and prior glycobiological knowledge. The approach presented is independent of target, prior knowledge, and anticipated glycan composition. Thus, this is a seminal work that would be beneficial for the glycobiology community. However, I have some concerns that I would like to be addressed before the manuscript is accepted for publication.

Major-

1. Although authors have shown structures of N-glycans with some linkages and some ambiguity, is it unclear if the automated method is being used to assign MS/MS? Was there a need to manually look at MS/MS spectra to confirm certain structures? This should be commented on. Also, some structures were left with ambiguity- which could be assigned manually- is it left unassigned because of the automated approach?

2. The author indicates MS spectra annotated with N-glycan structures, but hardly any MS/MS spectra (a few in the supplemental file) for these structures have been shown. It would be good to add some representative MS/MS spectra for some important and/or isomeric N-glycan structures for each tissue. 1-2 spectra could be added to the main paper, and the rest can go as a supplemental file. In addition, no Excel or other file is provided with a list of annotated N-glycan structures for each tissue with their respective relative abundances. The files on GitHub- contain abundances for all m/z, but it's unclear which of these belongs to N-glycans (and which N-glycan).

3. Also, the authors should comment on the analytical considerations of this work. How much-starting material was used in this study for the N-glycan release? Was it the whole tissue, and then how much of the approximate N-glycan was loaded on the column? (author provides this information as 5ul (of 20ul). It is not clear how much tissue/protein and then N-glycan is needed for the automated method. Could this work on lower amounts of protein with fewer N-glycans and low-intensity

TIC/EICs? What are the author's recommendations for minimum starting material and N-glycan to be loaded on the column?

4. It would be good to provide the readers with a table of all the samples used- male/female, age and type of tissue.

5. Although the authors mention the Symbol Nomenclature for Glycans "(SNFG) and explain the sub-structural determinants, unusual and isomeric N-glycans- but for non-glyco audiences – it would be good to provide the nomenclature for each of the glycan monosaccharide (e.g., red triangle for fucose) and then also show examples of the sub-structures, isomers, and unusual glycans discussed in the study. Also, a simplistic biosynthetic pathway with N-glycan structures and related enzymes, as discussed in the study, would assist non-glyco readers in understanding the work.

6. The authors have covered 20 different mouse brain tissues, but it would have been good to apply it to at least one similar human brain tissue (such as the brain) and see if the method gives similar results. This could have been achieved using publicly available data from human tissues using similar PGC-LC-MS/MS datasets to test the method's applicability and versatility.

Minor-

1. pg 23, line 639- female (f) of male (M)- relace 'of' by 'or'

2. Pg 5, line 28, remove an extra period.

Version 1:

Reviewer comments:

Reviewer #1

(Remarks to the Author)

I appreciate the addition of the benchmark analysis and manual annotation, which enhances the credibility of the results.

I understand that the main contribution of this work is the development of computational methods aimed at profiling classes of N-glycans to characterize the diversity and heterogeneity of the N-glycome across various mouse tissues, rather than focusing on the specific profiling of individual N-glycans. Given the challenges in accurately determining detailed glycan structures and the lack of comprehensive data on glycan isomer analysis, I would recommend that the title and the main text be revised to more accurately reflect the scope of the study. Specifically, using the term "non-targeted N-glycome profiling" or "profiling classes of N-glycans" instead of "isomer-sensitive analysis of glycan structures" would provide a more precise description of the research and prevent overstating the findings.

After the thorough revisions, I look forward to seeing this article published.

Reviewer #2

(Remarks to the Author)

The authors have thoroughly addressed my comments. I appreciate their efforts to compare the results from their oxonium-ion-based glycan identification algorithm on the mouse brain data. Originally I suggested to use synthetic glycan standard to perform the testing. But given the extra work to perform the experiment, I will not insist to add this experiment into the current manuscript. Perhaps the authors can consider this benchmark in their future work. I still have a little bit concern about the quantification method based on spectral counts, because the spectra counts may also be indirectly dependent on the responses of glycans in mass spectrometry. The authors may want to acknowledge the limitation of this semi-quantification method.

Reviewer #3

(Remarks to the Author)

The authors improved the manuscript as requested and made an appropriate revision. I recommend the revised manuscript for publication/.

Point-by-point reply

Reviewer #1 (Remarks to the Author):

This manuscript by Helm et al. presents a large N-glycome dataset comprising 20 different mouse tissues, generated through PGC-LC-MS/MS analysis and described a data analysis workflow for interpreting these data. The unbiased determination of glycans across different organisms has long been a necessity and a challenge. The substantial corpus of publicly available N-glycome data contributed by this study is valuable to the fields of glycomics and glycobiology.

We thank the reviewer for expressing his/her/their appreciation of the study's objective, and the immediate relevance of our dataset to the field!

However, the impact of this work has been diminished by the lack of rigorous and transparent data interpretation, thereby undermining the credibility of the conclusions drawn from these results. The primary concern is that the title of the manuscript suggests it presents an analysis method capable of distinguishing between glycan isomers. However, actually, only about 1/6 of the content discusses approach for discriminating glycan terminal monosaccharides, such as sialic acid, based on chromatographic retention times. This approach is not particularly novel and is commonly found in other glycomics analyses.

We deeply regret that the primary concerns of Reviewer 1 were prompted by - what we believe is – an unfortunate misunderstanding of both, the title, and the scope of our manuscript. In contrast to the reviewer's reading of its running title (i.e. “*Non-targeted isomer-sensitive N-glycome analysis reveals new layers of organ-specific diversity in mice*”) and as is stated in the abstract, our manuscript primarily aims at “[...] *make publicly available a consistent N-glycome dataset of 20 different mouse tissues and demonstrate a multimodal data analysis workflow [...]*”. In other words, in our manuscript, we wish to focus on new computational strategies towards the unbiased, non-targeted analysis of large-scale N-glycome datasets (i.e. “*highly scalable, LC-MS/MS data-driven method”*, from the abstract), and their respective results when applied to an extensive experimental dataset (i.e. 40 PGC-LC-MS/MS runs, analysing 20 different mouse tissues).

Furthermore, while it is certainly true that only a lesser part of the manuscript reports on the separation of glycan-isomers by PGC-chromatography, we do demonstrate throughout the manuscript multiple data-analytical approaches that integrate isomer-sensitive information other than chromatographic retention time. It is important to note, that glycan isomer-sensitive information is also provided by e.g. MS/MS derived diagnostic glycan-fragment ions. As examples, we show the identification of alpha-galactosylated structures (which are often isomeric to N-glycans without alpha-galactose), the discrimination between antennae- and core-fucosylated structures, the position of sialic acids within N-glycans (i.e. sialylated galactose vs sialylated GlcNac or polysialic acids vs the di-

sialyl Lewis C epitope), all exclusively based on the detection of sub-structure specific fragment ions in MS/MS.

In the light of this comment, we have now amended multiple sections of the main text (highlighted as tracked changes) to improve the clarity of the manuscript.

Moreover, the glycome analysis in this work involves direct extraction of unstructured information from mass spectrometry data without hierarchical processing or quality control. Consequently, it cannot serve as a comprehensive framework for glycan composition/structure-level mass spectrometry data analysis. Each step of data interpretation is conducted independently, lacking a systematic process for data retrieval and analysis.

We disagree with the reviewer. Throughout the manuscript, at different levels of raw-data processing and data-quality control, we demonstrate that exactly this “direct extraction of unstructured information from mass spectrometry data” does indeed provide meaningful and valuable information on the composition and/or structural features of the N-glycome.

In our manuscript, we wish to present multiple computational strategies (e.g. MS/MS-based N-glycome profiling, automated N-glycome reconstruction from raw LC-MS/MS data, eSNOG filtering for the identification of specific N-glycan sub-structural features) towards the unbiased, non-targeted analysis of large-scale N-glycome datasets, and apply them to one extensive dataset. We showcase and discuss the results of each of these independent approaches in the context of an extensive murine N-glycome dataset.

For instance, in the first section, "Precursor-independent MS/MS-based N-glycome profiling of 20 mouse tissues," the authors primarily extract information based on specific fragment ions, inferring differences among tissues based on ion intensities. However, such quantification is rudimentary and overlooks factors like fragmentation efficiency in mass spectrometry.

We thank the reviewer for pointing out this unclarity to the reader! We do agree, that this approach is fairly simplistic and merely provides an estimate for the abundance of glycans or glycan-types. Conceptually, our MS/MS-based N-glycome profiling approach is similar to “spectral counting”, a well-established semi-quantitative data-analysis approach in the fields of proteomics. It does not take fragment ion intensity (other than the respective ion having been detected in a specific MS/MS spectrum, or not) into account. Instead, the semi-quantitative information derives from the number of MS/MS spectra that do contain signals of glycan-feature specific fragment ions, divided by the number of all MS/MS spectra generated during one LC-MS/MS run that contain the fragment ion of 224.1 amu. We now amended the respective section with additional clarifications, definitions and references.

The filtering method used in this work based on SNOG-scoring and eSNOG-scoring relies on empirical score. It's worrisome whether this filtering method is universally applicable to all glycome data, given the significant variations in glycomics analysis.

This is true! We thank the reviewer for pointing out this important limitation of this approach, which we did not discuss and highlight sufficiently well in the previous version of the manuscript. We now highlight and discuss this critical feature of the SNOG-score function in the respective section:

“In contrast to our initial, binary spectral filtering approach, SNOG-scores incorporate critical information on fragment ion intensities. While this makes SNOG-score based MS/MS data filtering generally more robust against e.g. co-isolation events, it is important to note that it also renders SNOG-score values dependent on the actual experimental MS/MS acquisition parameters (e.g. collision energy settings). For our dataset (i.e. generated using stepped HCD collision energies 20, 25 and 30%), we empirically determined that MS/MS spectra with SNOG-scores greater than 0.03 derive from actual N-glycan precursors (Supplementary Fig. 3., Supplementary Fig. 4., and Supplementary Fig. 5.). Other experimental parameter settings may require different SNOG-score value thresholds for effective data-filtering.”

Other concerns include:

- 1) The authors stated that each sample was analyzed in duplicates, but when conducting statistical analysis of different glycan profiles among tissues, such as in Fig.1, SFig.1, and SFig.2, the differences and significance were not statistically analyzed.*

In the light of good statistical practice, we prefer to not apply the proposed statistical methods to such small sample groups (i.e. duplicates). Instead, we now show the individual duplicate data points in Figures 1.,3.,4., and 5..

- 2) The authors employed the diagnostic ion for reduced N-acetylhexosamine to filter spectra, obtaining 220,506 spectra from 509,283 spectra. Was the intensity threshold of 224m/z considered during filtration?*

No. At this step (i.e. MS/MS-based N-glycome profiling) we did not apply any fragment ion intensity threshold as a filter criterion. The approach does not take fragment ion intensity (other than the respective ion having been detected in a specific MS/MS spectrum, or not) into account. Instead, the semi-quantitative information derives from the number of MS/MS spectra that do contain signals of specific glycan-related fragment ions divided by the number of all MS/MS spectra generated during one LC-MS/MS run that contain the fragment ion of 224.1 amu).

We now amended the respective section with additional (also conceptual) clarifications, definitions and references:

“For this, we automatically extracted all MS/MS spectra and retained only those that contained N-glycan specific fragment ions (i.e. 224.1 amu, diagnostic for reduced N-acetylhexosamine). From this data, we generated semi-quantitative information on the MS/MS level by implementing a spectral counting approach, similar to methods used in the fields of proteomics^{31,32} More specifically, we developed a binary MS/MS spectrum counting approach, that is precursor-mass and -intensity independent and exclusively based on the detection (or complete lack) of N-glycan specific fragment ions.”

Additionally, were the filtered-out spectra containing other glycan fragments?

Yes. A small fraction of the filtered-out MS/MS spectra contained other glycan fragments. Manual inspection of the rejected MS/MS spectra (i.e. MS/MS spectra that did not exhibit the 224.1 amu fragment ion) showed that a big share of these spectra did not contain any other known glycan fragments and were thus deemed to having been derived from non-glycan contaminants (i.e. represent correctly filtered out by our script). Only a small number of filtered-out MS/MS spectra did include fragment ions derived from glycans. The lack of the 224.1 amu fragment ion indicates that these spectra were most likely derived either from in-source fragments of reduced N-glycans, from non-reduced N-glycans, or from other contaminant glycans (e.g. glycogen in liver samples) without a reduced HexNAc-residue (i.e. also these spectra were correctly filtered out by our script). We now added this information to the respective section of the manuscript:

“Manual inspection of rejected MS/MS spectra (i.e. MS/MS spectra that did not exhibit the 224.1 amu fragment ion) showed that a big share of these spectra did not contain any other known glycan fragments either and were thus deemed to having been derived from non-glycan contaminants (and therefore correctly filtered out by our script).”

The authors utilized glycan diagnostic ions to showcase the abundance of a particular glycan type within a sample. However, the sources of glycan fragments for a specific type are not singular. Taking Neu5Ac-containing glycans as an example, while it's accurate to consider 292 Da and 274 Da, there could also be fragments like 495 Da. How did the authors calculate the proportion of Neu5Ac-containing glycans in a sample?

We do agree with the reviewer that there may be other fragment ions in MS/MS spectra that derive from Neu5Ac-bearing glycans. Still, to determine the general presence (and estimate abundance) of Neu5Ac in an N-glycome dataset we exclusively rely on the 292 and 274 amu fragment ions. Larger Neu5Ac-related fragment ions (such as Neu5Ac + Hex, 454 amu, or Neu5Ac + HexNAc, 495 amu) encode structural features, which are not imperative for all Neu5Ac-containing N-glycans. The semi-quantitative information provided in this section of the manuscript derives from the number of all MS/MS spectra that do contain signals of the fragment ions 224, 292 and 274 amu, divided by the number of all MS/MS spectra generated during one LC-MS/MS run that contain signals of the fragment ion of 224 amu.

Larger diagnostic fragment ions were only used to further stratify the Neu5Ac-bearing N-glycan populations in later sections of the manuscript, in conjunction with the eSNOG function (e.g. we used 495 Da to identify N-glycans carrying a sialylated HexNAc or 948 Da to identify the sialyl Lewis C epitope).

- 3) *Utilizing PGC retention times for isomer analysis isn't a novel method. What specific optimizations did the authors make regarding PGC? What advantages does your approach offer for isomer analysis compared to existing methods? Regrettably, in this manuscript, apart from applicability to larger datasets, I didn't see any other advancements.*

We agree that our manuscript does not report on improvements or optimizations of PGC-chromatography. However, this is not the scope of the present work. Instead, we wish to present computational strategies that are scalable and applicable to even extensively large N-glycome raw-datasets. We did not aim at improving or optimizing PGC.

- 4) *The authors mentioned that their analysis can address issues such as incorrect mono-isotopic peak-picking or charge-state assignments by the mass-spectrometer, unintended precursor ion co-isolation, in-source fragmentation, or excessive adduct-ion formation (Line189-191). However, they didn't provide corresponding evidence through benchmark data or comparisons to demonstrate how these issues were addressed or the extent of improvement achieved.*

To corroborate our statements, we now provide an extensive and detailed example for the impact of our computational strategies on LC-MS/MS data quality. For this, we iterated through the critical data-processing steps involved in the analysis of our mouse liver dataset and quantified the impact of each step.

Conceptionally, all our computational approaches presented in this manuscript put "MS/MS data first". As our study shows, these strategies allow for the untargeted analysis of N-glycan precursors, without prior knowledge of glycan compositions and/or masses for a given sample. Importantly, this also extends to the identification of unexpected glycan precursor masses, including adduct ion

masses, which are readily identified from MS/MS data (irrespective of their actual precursor mass).

In turn, this “MS/MS data first” paradigm makes our approaches critically dependant on highly confident MS/MS precursor information as part of our dataset. It is therefore important to note, that the “native” MS/MS precursor information (i.e. mono-isotopic precursor m/z and charge-state information; which is generated by the generic mass-spectrometer data-acquisition-software) has been found highly defective for decades, which prompted the development of numerous data-processing/correction tools (e.g. DOI: 10.1021/acs.jproteome.0c00563, DOI: 10.1002/pmic.201100081, DOI: 10.1021/jasms.2c00176).

To compensate for defective, “native” MS information, in our study, we employed the commercial proteomics search-engine software-suite PEAKS for MS/MS data-refinement. To the best of our knowledge, our study presents the first example of MS/MS precursor refinement in the context of glycome analysis. The data-refinement steps which were implemented in PEAKS, re-evaluate both parameters, the MS/MS precursor charge-state and the mono-isotopic peak assignment.

To evaluate the extent and impact of this critical data-processing step, we compared PEAKS-processed data to those that were extracted using the program msConvert (part of the ProteoWizard software package; DOI: 10.1007/978-1-4939-6747-6_23). Most importantly, in contrast to PEAKS, msConvert does not correct and puts out the “native” (i.e. mass-spectrometer derived) MS/MS precursor information.

Comparison of the two MS/MS datasets (both generated from the LC-MS/MS raw file “mouseNglycome_liver_1.raw” and put out in the .mgf file format;), revealed important disagreement between the “refined” and the “native” MS/MS dataset. More specifically, PEAKS data-refinement yielded a different MS/MS precursor ion charge-state for 30.2% of all MS/MS spectra (i.e. 5016 of 14913; Rev. Fig. 1.A) and led to the reassignment of the mono-isotopic MS/MS precursor mass for at least 21.4% of all MS/MS spectra (i.e. 3206 MS/MS scans; Rev. Fig. 1.B). The “native” precursor information of 65.2% of all MS/MS spectra was changed by PEAKS. Manual inspection of the raw data-file (e.g. Rev. Fig. 1.C) confirmed many data-conflicts correctly resolved by PEAKS. A final statement over the fidelity of the data-refinement by PEAKS, however, would require the manual curation of at least all 14193 MS/MS scans in the dataset, which is clearly beyond the scope of the present study.

Importantly, however, the striking differences in the MS/MS data-foundation generated by these two data-extraction methods (i.e. depending on raw-data extraction and refinement software used) also have repercussions on the subsequent precursor mass-bin selection (Rev. Fig. 1. D), N-glycan precursor

filtering (Rev. Fig. 1. E,F) and the reconstruction of semi-quantitative precursor mass-bin histograms (Rev. Fig. 1. G). For example, only 76.8% of all PEAKS derived and SNOG-filtered precursor mass-bins conformed with 71.3% mass-bins obtained from the msConvert dataset. In a next step, we thus extracted precursor-mass bin specific signal intensities from the Decon2 processed LC-MS raw data and aligned them with the N-glycan precursor mass-bins identified above. Remarkably, the SNOG-filtered mass-bins from both datasets covered each approx. 20% of the summed signal intensity recorded in the experiment raw-data (Rev. Fig. 1. G), with an overlap of 98% of summed signal intensity between PEAKS and msConvert processed data. This observation suggests that most discrepancies between the two datasets relate to differential processing of low abundant precursor signals and indicates that SNOG filtering robustly compensates for inaccuracies in the assignment of “native” precursor information on low-abundant analytes.

By contrast, the identification of N-glycan precursor mass-bins from “native” MS/MS precursor information by filtering for the mere presence of diagnostic fragment ions (e.g. intensity of MS/MS fragment mass 224.1 amu > 0) resulted in 2482 precursor masses, that covered more than 62% of all MS signals recorded (Rev. Fig. 1. H). Comparison and close inspection of the reconstructed precursor mass-bin histograms highlights substantial differences in the identification of a series of mass-bins, not detected in SNOG filtered data (Rev. Fig. 1. I). From manual inspection of representative MS/MS scans, and as was already discussed in the main text of the original submission and shown in SI Figure 5., we showed that these mass-bins do not contain N-glycan precursors, but rather represent oligo-hexose polymers (e.g. glycogen fragments). However, due to unintentional co-isolation of low-abundant N-glycan precursors (or in-source fragments or adducts thereof) for MS/MS, some spectra do indeed contain minimal signals for the “N-glycan diagnostic” fragment ion 224.1 amu and were thus not rejected by this simplified filtering approach. Our SNOG scoring function, on the other hand, incorporates a fragment ion intensity threshold, which effectively filters these spectra out.

Taken together, this showcase of our computational approach, comprising raw data refinement, data-aggregation into precursor mass-bins, the application of an MS signal intensity threshold and SNOG-score based filtering, effectively identifies N-glycan precursor masses from complex samples in an automated, non-targeted fashion and markedly reduces the impact of non-N-glycan contaminants, unintended precursor co-isolation and/or in-source fragments.

Rev. Fig.1

Rev. Fig.1 – Showcase of computational approach to improve MS/MS raw-data quality. A) Differential analysis of the results of two raw-data extraction algorithms (msConvert, PEAKS) reveals potentially defective, “native” MS/MS precursor charge state information. Data-points indicate the charge-state corrected $M+H^+$ precursor mass of all individual MS/MS scans ($n=14913$ MS/MS scans) that were extracted from the raw file “mouseNglycome_liver_1.raw”. Data point size represents the intensity values of the N-glycan related fragment ion 224.1 amu in each MS/MS scan). Differences in

precursor charge state information between MS/MS scan populations are indicated by numbers. **B)** Quantification of alternative mono-isotopic precursor masses between msConvert and PEAKS, in the mass-range of -5 to 5 amu. **C)** raw data example of MS/MS scan with different precursor information after msConvert (charge-state =2+) or PEAKS (charge-state = 1+) processing. Raw MS data generated at the chromatographic time-point (upper panel) of the specific MS/MS scan (scan number 5631; lower panel) corroborates correct precursor charge-state detection by PEAKS. **D)** Venn-diagram of all precursor mass-bins (bin width = 0.1 amu) constructed from either msConvert or PEAKS processed data. **E)** Venn-diagram of all precursor mass-bins (bin width = 0.1 amu) constructed from SNOG-filtered and PEAKS processed MS/MS spectra versus msConvert processed data filtered for the mere detection of fragment ion 224.1 amu. **F)** Venn-diagram of all precursor mass-bins (bin width = 0.1 amu) constructed from SNOG-filtered and PEAKS processed MS/MS spectra versus SNOG-filtered and msConvert processed data. **G)** Histogram of summed Decon2-intensity values for all mass-bins in the mass-range from 500 – 4000 amu (grey), for mass-bins derived from SNOG-filtered PEAKS data (green) and for mass-bins derived from SNOG-filtered msConvert data (blue).

- 5) *In this study, the authors calculated that almost 50% of the brain N-glycome is sialylated. It's noteworthy that in other studies, proportions of 3%, 20%, and 40% have been reported (line 323). What factors could lead to such significant differences? Was it due to previous methods lacking sensitivity, experiencing in-source fragmentation, or difficulty in detecting sialic acid? The author not only failed to present a detailed comparison of methods but also didn't engage in thorough discussion, attributing the differences solely to methodological variations, which is insufficiently rigorous. How should one choose a method when they want to research the content of specific glycosylation?*

The degree of sialylation reported for the murine brain varies vastly between studies, with previous estimates ranging from only ~3% to 40% of sialylation. From revisiting the respective publications, we think that these remarkable discrepancies may indeed be best attributed to “methodological variation” (i.e. multiple differences in sample quality, glycoprotein extraction, N-glycan preparation and analysis), and uphold our summative statement.

For example, Williams et al. analysed permethylated N-glycans from 4 different brain sections (i.e. cortex, hippocampus, striatum, cerebellum) by MALDI-MS. Barboza et al. analysed non-reduced native N-glycans from brain cell-membrane fractions of 5 different brain sections (i.e. forebrain, hindbrain, cortex, hippocampus, and cerebellum) by nanoChip-PGC-LC-MS(/MS).

Furthermore, in their work, Williams et al. lysed the respective tissues, digested the protein extracts with trypsin, recovered PNGase F released N-glycans from the flow-through of C18 SepPak SPE cartridges and permethylated them (i.e. subjected to another multi-step sample-processing procedure!). Barboza et al., on the other hand, extracted region-specific brain cell membrane fractions by ultracentrifugation, released N-glycans from denatured (but not tryptically digested!) membrane-protein extracts, recovered them using porous graphitized carbon (PGC) SPE cartridges and reduced them using borohydride (and presumably engaged in an additional round of sample clean-up to recover the reduced N-glycans prior to analysis).

Next to the specific instrumental platforms and MS-analytical methods, sample-preparation techniques have been reported to impact specifically on the recovery and detection of sialyated N-glycan species. Most recently, for example, Moh et al. (DOI: 10.1021/acs.analchem.3c04928) used negative mode PGC-LC-MS/MS for the analysis of reduced N-glycans from whole mouse brain, comparing two sample preparation methods side-by-side, and yielded approx. 22% and 32% of sialyated N-glycans, depending on the sample preparation method they used.

Although we agree with the reviewer, that a detailed review of and discussion on methodological differences in N-glycome studies (particularly between these two seminal studies) would be interesting, we do consider it beyond the scope of the present manuscript.

- 6) *There are minor changes needed as well, such as replacing "amu" with "Da," which is more commonly used in current mass spectrometry analysis, or "m/z." However, this change is not imperative.*

We thank the reviewer for this suggestion.

Reviewer #1 (Remarks on code availability):

The R script, MS1 and MS2 files, along with a README file, are provided. The code works, but I haven't run it on all of the data.

Reviewer #2 (Remarks to the Author):

The paper aims to characterize the diversity and heterogeneity of N-glycome across various tissues in mouse. Instead of the profiling of specific N-glycans, the paper developed computational methods for profiling classes of N-glycans (e.g., sialylated, sulfated and fucosylated glycans, or the glycans with different linkage of sialic acids). The profiling methods are based on the MS/MS spectra of glycans containing signature oxonium-ions for different classes, which is easy to be implemented and applied. The paper also presented the relative abundance profiles of isomeric peaks based on the retention profiles under LC. The analyses across 20 mouse tissues revealed specific glycan patterns in different tissues (e.g., in brain vs. other tissues).

The computational methods used in the study is simple but interesting. Using signature oxonium-ions to classify MS/MS spectra of N-glycans have been long used, in particular to filter glycan spectra in high-throughput analyses. This seems to a comprehensive study based on this approach leading to interesting findings.

We thank the reviewer for his/her/their generally positive assessment!

However, the accuracy of the method has not been thoroughly validated previously. So it is useful for the authors to first establish a baseline of potential errors of such approach. In particular, the intensities of oxonium-ions appears to be highly correlated with the experimental condition such as fragmentation energy.

This is true. We thank the reviewer for pointing out this important limitation of this approach, which we did not discuss and highlight sufficiently well in the previous version of the manuscript. We now highlight and discuss this critical feature of the SNOG-score function in the respective section:

“In contrast to our initial, binary spectral filtering approach, SNOG-scores incorporate critical information on fragment ion intensities. While this makes SNOG-score based MS/MS data filtering generally more robust against e.g. co-isolation events, it is important to note that it also renders SNOG-score values dependent on the actual experimental MS/MS acquisition parameters (e.g. collision energy settings). For our dataset (i.e. generated using stepped HCD collision energies 20, 25 and 30%), we empirically determined that MS/MS spectra with SNOG-scores greater than 0.03 derive from actual N-glycan precursors (Supplementary Fig. 3., Supplementary Fig. 4., and Supplementary Fig. 5.). Other experimental parameter settings may require different SNOG-score value thresholds for effective data-filtering.”

It is not uncommon that some MS/MS spectra of glycans miss certain oxonium-ions. It is unclear if the classification based solely on signature oxonium-ions (i.e., the SNOG-score) will result in inaccurate estimation of different classes of N-glycans in the complex sample. It would

be useful to test this method using standard samples to estimate the level of variance across samples and across N-glycan classes due to inaccurate estimation.

In lack of a publicly available and annotated dataset, that would allow to directly benchmark our computational approach, we now compared our results on mouse brain, to those presented in a very recent PGC-LC-MS/MS dataset of reduced N-glycans from whole mouse brain by Moh et al. 2024 (DOI: 10.1021/acs.analchem.3c04928). It is important to note, that the dataset by Moh et al., was generated using an ion-trap instrument (i.e. LTQ Velos Pro) operated in negative mode. Our data were generated by an Orbitrap instrument, operated in positive ion-mode. This methodological difference implies that quantitative differences between specific glycan classes may be expected due to varying ionization efficiencies of negatively charged N-glycans. Classification of N-glycans by our approach, however, is expected to confirm with the annotation by Moh. et al.

For initial alignment of the two datasets (Rev.Fig.2. A,B), the theoretical mass values of N-glycan precursors (as were provided by Moh et al.) were again transformed from [M-H]⁻ to [M+H]⁺, assigned to precursor mass-bins (of bin-width = 0.1 amu) and paired with the respective summed "peak area" (as were provided by Moh et al.). This dataset was then compared to the mouse brain data that we presented in our manuscript.

Qualitative comparison of the two independent datasets confirmed excellent agreement between 90 N-glycan precursor mass-bins that were identified in both datasets and covered 95.8% of the total peak area originally annotated by Moh et al., or 71.2% of the SNOG-scored total ion-intensity presented in our study (Fig. Rev2.D). Only 13 (out of the 103) mass-bins annotated by Moh et al., were not immediately congruent with our dataset (Rev.Fig.2. C,D). Of those 13, 7 precursors were only detected in one of our two analytical runs (i.e.1641.5, 2954.1, 3069.1, 3174.1, 3272.2, 3377.2, 3597.3) and were thus excluded in from our comparison. Three other precursor mismatches (i.e. 1869.6, 2930.0, 3434.2) related to inconsistent digit-rounding between the two datasets, which resulted in their signals having been allocated to neighbouring mass-bins (e.g. 1869.6 → 1869.7) and having been well detected by our approach after all. Only two precursors (i.e. 2031.6, 2564.9), annotated as doubly sulphated or triply Neu5Gc-sialylated N-glycans by Moh et al., respectively, were not detected in our study whatsoever.

Manual inspection of the 90 N-glycan precursor mass-bins that are shared between both datasets further corroborated close to perfect agreement. For example, 9 of the 13 of Neu5Ac-containing di-sialyl Lewis C-epitope N-glycan masses in our study (main-Fig. 5.B) were identified with the exact same monosaccharide compositions by Moh et al., yet not structurally assigned to contain the di-sialyl Lewis C-epitope. Instead, Moh et al. annotated the respective structures with ambiguity in Neu5Ac linkages.

In turn, at least 5 mass-bins (e.g. 1155.4, HexNAc2Hex4ph1; 1383.5, HexNAc2Hex5Fuc1; 2242.8, HexNAc4Hex6Fuc1Neu5Ac1) that were uniquely identified in our dataset, could also be manually extracted and tentatively assigned by us from the LC-MS/MS raw data provided by Moh et al. (Rev.Fig.3. A-E).

Quantitatively, however, the alignment of two independent (and methodologically very different) N-glycome studies shows marked differences regarding precursor intensity values reported (Rev.Fig.2. C). Most importantly, in comparison to our data, Moh et al. reported substantially higher levels of oligo-mannose N-glycan structures (i.e. HexNAc2Hex6, HexNAc2Hex7, HexNAc2Hex8 and HexNAc2Hex9, Rev.Fig.2.C) in their mouse brain samples - using the SSSMuG sample preparation protocol. Importantly, these quantitative differences also impact on the calculation of e.g. the degree of sialylation. Again, we hypothesise that these quantitative differences are essentially due to methodological variation between the two studies.

Rev. Fig.2

Rev. Fig.2 – Benchmark mouse brain N-glycome analysis. **A)** Alignment of precursor intensity histograms showing peaks-areas as reported by Moh *et al.* 2024 (blue) and SNOG-filtered, Decon2 extracted TIC fractions reported on this study (red), in the mass-range from 500 to 4000 amu. Raw Decon2 extracted TIC (i.e. of all precursor mass-bins) is depicted in grey. **B)** Differential analysis of N-glycan precursor mass-bins identified by Moh *et al.* 2024 (blue) and this study (red). **C)** Zoom-view of precursor intensity alignment in A, in the mass-range from 500 to 4000 amu. Uniquely identified N-glycan compositions and theoretical [M+H]⁺ masses are indicated in red. Precursor mass-bins identified in both studies are indicated in black. **D)** List of precursor mass bins identified by Moh *et al.*, not immediately congruent with this study. Two mass bins that were -even upon manual inspection of the raw data- not detected in this study are highlighted in bold.

Rev. Fig.3

Rev. Fig.3 – Manual re-analysis of raw-data by Moh et al. 2024 for N-glycan masses uniquely identified in our study. A-E) examples of MS (left panels) and MS/MS data of N-glycan precursors that were uniquely detected in our study, not annotated by Moh et al. 2024, and yet found in the manual re-analysis of the raw data of Moh et al. 2024. Our proposed glycan compositions and their theoretical [M+H]⁺ masses are indicated as red inserts.

Furthermore, we specifically compared the fraction of Neu5Ac sialyated N-glycan species identified in the two studies. For this, we extracted the manual annotation provided by Moh et al. for all Neu5Ac-containing structures and aligned them with our eSNOG274/292 filtered N-glycome data of the mouse brain. Again, qualitatively, we do find good agreement 44 N-glycan precursor mass-bins that were identified in both datasets and covered 83.7% of the Neu5Ac-associated peak area originally annotated by Moh et al., or 59.3% of the eSNOG274-scored total ion-intensity recorded in our study, respectively (Rev.Fig.4.A), but do see varying quantitative differences with regard to the abundance of specific Neu5Ac-containing N-glycan precursors (Rev.Fig.4.B).

Rev. Fig.4

Rev. Fig.4 – Sialome comparison. **A)** Differential analysis of the Neu5Ac containing fraction of the mouse N-glycome of this study (red) and Moh et al. (blue). **B).** Zoom-view into the precursor intensity value histogram of Neu5Ac containing mass-bins of this study (red; Decon2 extracted TIC values) and Moh et al. (blue; peak area values) in the mass-range from 2000 – 2500 amu, confirms good qualitative congruence, and indicates quantitative differences for specific N-glycan structures.

Similar quantitative differences (without clear biases towards specific N-glycan compositions or structures) between our data and other N-glycome studies became apparent when benchmarking our results to e.g. those of mouse kidney and mouse lung by Shubhakar et al., 2018 (DOI: 10.1007/s10719-018-9825-8), a MALDI-MS analysis of permethylated N-glycans, quantitatively (Rev. Fig. 5.A,B).

Rev. Fig.5

Rev. Fig.5 – Quantitative comparison. **A)** mouse lung N-glycans identified and quantified by Shubhakar, et al. 2018(blue) and this study (green and dark green). **B)** kidney N-glycans identified by and quantified Shubhakar, et al. 2018(blue) and this study (green and dark green).

Taken together, our data comparison shows that our computational approach, comprising raw data refinement, data-aggregation into precursor mass-bins and the application of SNOG-score based filtering, effectively identifies N-glycan precursor masses from complex samples, and that both, our N-glycome data and our results, as well as N-glycan stratification, largely conform with those obtained from comparable, manually annotated datasets.

Some methods were not completely clear from the methodology description. For examples, how are the glycan abundance profiles in Fig 1 estimated? Are they based on the total intensity of the precursor ions (in MS1) classified in each class of N-glycans? If that is case, does it assume different classes of N-glycans (e.g., sialylated vs. others) have similar responses in MS?

We thank the reviewer for pointing out this unclarity to the reader! We agree that different N-glycan classes have different responses in MS. Our MS/MS-based N-glycome profiling approach is similar to “spectral counting”, a well-established semi-quantitative data-analysis approach in the fields of proteomics. It does not take the precursor ion intensities into account. Instead, the semi-quantitative information presented in Fig.1. derives from the number of MS/MS spectra that do contain specific signals of glycan-related fragment ions divided by the number of all N-glycan related MS/MS spectra (i.e. containing fragment ion mass 224.1 amu with intensity > 0) generated during one LC-MS/MS run. We now amended the respective section:

“To dissect this important compositional heterogeneity of N-glycans, we extended our MS/MS data-filtering criteria to sialylation-specific diagnostic fragment ions (i.e. reduced HexNAc fragment ion mass 224.1 amu and Neu5Ac fragment ion mass 292.1 amu and/or Neu5Gc fragment ion mass 308.1 amu) and counted the associated MS/MS spectra. To ensure that only spectra derived from reduced N-glycans are included, all spectra without the 224.1 amu fragment ion were excluded before the counting step.”

Also, How are the isomeric glycans with different linkages identified in the retention profile (e.g., in Fig. 6).

Identification of glycan structures was performed manually and is based on the elution time points of N-glycan structures, which we could safely identify with the help of other studies, in conjunction with previous knowledge on the retention properties of the specific N-glycan features. For example, it is known that structures with a bisecting GlcNAc elute earlier than the same structure with the GlcNAc on different position. Furthermore, previously established PGC-elution rules, standard libraries, and MS/MS information were used in this regard. We now added this information to multiple sections of the manuscript.

Finally, it is unclear if and how the analysis were validated through biological/technical replicates. In ln 233-234, it is stated that "... validated robust reproducibility among sample replicates ...", but no details were given.

We agree that this statement might be misleading! We apologize for not having been clear enough and thank the reviewer for this comment. We tried to highlight good reproducibility between biological samples, as is shown by the results of correlation and hierarchical cluster analysis depicted in Fig. 3. We modified the respective section of the manuscript accordingly for improved clarity:

“The correlation and cluster analysis of our SNOG-filtered dataset (Fig. 3.) revealed distinctive tissue-specific clustering patterns.”

and:

“The measured samples derived from different animals and were individually processed. The yet high degree of similarity across all tissue pairs evidences a high technical reproducibility of our analyses (Fig 3.).”

Reviewer #2 (Remarks on code availability):

The code is relatively simple and works fine.

Reviewer #3 (Remarks to the Author):

The presented manuscript by Helm et al. provides a consistent N-glycome dataset of 20 different mouse tissues and demonstrates a multimodal data analysis workflow that allows for outstanding depth and coverage of N-glycome features. This highly scalable, LC-MS/MS data-driven method integrates the automated identification of N-glycan spectra, the application of non-targeted N-glycome profiling strategies, and the isomer-sensitive analysis of glycan structures. In addition, the manuscript also provided N-glycome features for critical sub-structural determinants and glycan isomers across the mouse N-glycome, uncovering tissue-specific glycosylation patterns and highlighting multiple layers of N-glycome complexity that derive from organ-specific regulations of glycobiological pathways. Overall, the manuscript is well-written and has compelling data. This is a vast data mining effort consolidating 20 different mouse tissue N-glycome profiling and provides automated identification and annotation of N-glycan spectra as opposed to conventional PGC-LC-MS/MS manual analysis that relies on the targeted, selective extraction for known or anticipated glycan compositions and masses, and subsequent structure assignment using pre-established, relative chromatographic retention times, complementary MS/MS data analysis, and prior glycobiological knowledge. The approach presented is independent of target, prior knowledge, and anticipated glycan composition. Thus, this is a seminal work that would be beneficial for the glycobiology community.

We thank the reviewer for his/her/their positive assessment!

However, I have some concerns that I would like to be addressed before the manuscript is accepted for publication.

Major- 1. Although authors have shown structures of N-glycans with some linkages and some ambiguity, is it unclear if the automated method is being used to assign MS/MS? Was there a need to manually look at MS/MS spectra to confirm certain structures? This should be commented on. Also, some structures were left with ambiguity- which could be assigned manually- is it left unassigned because of the automated approach?

Identification of glycan structures was performed manually and is based on the elution time points of N-glycan structures, which we could safely identify with the help of other studies, in conjunction with previous knowledge on the retention properties of the specific N-glycan features. It is correct, that -in our manuscript- many structures were left with ambiguity. This ambiguity often stems from uncertainties over the exact location of certain modifications, e.g. the arm location or its exact linkages type or position. This ambiguity is, in most cases, not possible to solve with MS/MS alone. We now added this information to multiple sections of the manuscript (indicated by tracked changes).

2. The author indicates MS spectra annotated with N-glycan structures, but hardly any MS/MS spectra (a few in the supplemental file) for these structures have been shown. It would be good

to add some representative MS/MS spectra for some important and/or isomeric N-glycan structures for each tissue. 1-2 spectra could be added to the main paper, and the rest can go as a supplemental file.

We thank the reviewer for this suggestion. We now added MS/MS spectra of representative MS/MS spectra of important N-glycan structures to Figs.4 and 5.

In addition, no Excel or other file is provided with a list of annotated N-glycan structures for each tissue with their respective relative abundances. The files on GitHub- contain abundances for all m/z, but it's unclear which of these belongs to N-glycans (and which N-glycan).

We now updated the raw SNOG filtered data-files (as .csv files) on GitHub. These files contain all SNOG-filtered (i.e. N-glycan) precursor mass-bins, with their respective intensity values, tissue specifically. Furthermore, these files contain diagnostic fragment ion abundance values of the SNOG-filtered precursor mass-bins, that can be used to further stratify the respective N-glycome dataset. N-glycan identification was performed manually. We now amended multiple sections of the manuscript to improve clarity to the reader.

3. Also, the authors should comment on the analytical considerations of this work. How much starting material was used in this study for the N-glycan release? Was it the whole tissue, and then how much of the approximate N-glycan was loaded on the column? (author provides this information as 5ul (of 20ul). It is not clear how much tissue/protein and then N-glycan is needed for the automated method. Could this work on lower amounts of protein with fewer N-glycans and low-intensity TIC/EICs? What are the author's recommendations for minimum starting material and N-glycan to be loaded on the column?

Our computational approach works largely independent of N-glycan input, provided that the analytes of interest give rise to MS- and MS/MS spectra with minimal data quality. Provided that our orbitrap, positive-mode dataset has been generated using a “capillaryLC”-typical flowrate (i.e. of 6 uL/min; rather than more widely used nano-LC flow rates in the range of 250 to 500 nL/min), all of which does not represent the predominant analytical platform for N-glycan analysis in the field, and the wide variations in input material for the respective tissues (e.g. kidney and brain yielded substantially more N-glycans than lymph nodes or skin), we prefer to not give recommendations on minimum input material. For reference of the performance of our computational workflow on lower input samples, analysed on a different type of mass-spectrometer, operated in opposite polarity, please find our showcase below.

4. It would be good to provide the readers with a table of all the samples used- male/female, age and type of tissue.

We agree and now include an additional table to the Supplementary Information.

5. Although the authors mention the Symbol Nomenclature for Glycans "(SNFG) and explain the sub-structural determinants, unusual and isomeric N-glycans- but for non-glyco audiences – it would be good to provide the nomenclature for each of the glycan monosaccharide (e.g., red triangle for fucose) and then also show examples of the sub-structures, isomers, and unusual glycans discussed in the study. Also, a simplistic biosynthetic pathway with N-glycan structures and related enzymes, as discussed in the study, would assist non-glyco readers in understanding the work.

We thank the reviewer for this suggestion. We agree that figures summarizing the discussed biosynthetic processes will help non-experts to understand the manuscript. We have therefore now included biosynthetic figure panels in Fig.2, Fig.5 and Fig.6.

6. The authors have covered 20 different mouse brain tissues, but it would have been good to apply it to at least one similar human brain tissue (such as the brain) and see if the method gives similar results. This could have been achieved using publicly available data from human tissues using similar PGC-LC-MS/MS datasets to test the method's applicability and versatility.

We agree that testing our computational approach on a publicly available, benchmarked datasets would be ideal. For this, we extensively reviewed essentially all glycan data that are publicly available on ProteomeXchange and the GlycoPOST repository, in search of datasets that were directly compatible with our computational approach. From this, we had hoped to identify MS/MS datasets that cover diagnostic oxonium ions in the low m/z range (e.g. 204.1, 224.1 amu; or related fragment masses that derive from permethylated N-glycans). Unfortunately, and to our surprise, we could not identify such a dataset.

In lack of a publicly available dataset that would allow to directly demonstrate the versatility of our computational approach, we nevertheless applied our computational approach to a very recent PGC-LC-MS/MS dataset of reduced N-glycans from whole mouse brain by Moh et al. 2024 (DOI: 10.1021/acs.analchem.3c04928). It is important to note, that this dataset was generated using an ion-trap instrument (i.e. LTQ Velos Pro) operated in negative mode. These methodological differences imply that raw data are fundamentally different to our Orbitrap positive ion-mode dataset: ion-trap instruments are less precise in mass on both, the MS- and the MS/MS-level, and fragmentation of negatively charged N-glycan precursors induces an entirely different set of glycan fragments in MS/MS. Most critical to our approach, in comparison to our dataset, the acquisition of negative mode MS/MS data on an ion-trap instrument suffers from a lack of diagnostic oxonium ions in the low m/z range. The paucity of these important fragments in the data clearly makes this a sub-optimal showcase and required several adaptations of our data-analytical scripts.

Nevertheless, for our showcase, we first transformed the MS/MS raw-data of 6 raw-data files of Moh et al. into the .mgf file format (using msConvert), extracted the intensity values of a set of generic negative-ion mode specific fragment ions that are diagnostic for the reduced end of N-glycans (i.e. 204.1, 222.1, 368.2, 407.2, 425.2, 553.2, 571.2; mass-precision = +/- 0.3 amu; DOI: 10.1007/s13361-013-0610-4), and divided their sum by the intensity value of the most intense fragment ion in the respective MS/MS spectrum (i.e. base-peak intensity; BPI), respectively. Based on these MS/MS spectrum-specific values, that were calculated from our negative-mode-specific adaptations to the SNOG-score function, we identified 55 N-glycan derived precursor mass-bins (bin-width of 0.5 amu; compensating for lower mass-precision of ion-trap raw-data) with a score of > 0. More than half of these precursor mass-bins (i.e. 31 of 55) were also manually annotated by Moh et al. (Rev.Fig.6. A), confirming that -despite the inadequacy of the MS/MS raw data and without any threshold optimization- our computational approach does indeed identify N-glycan precursor mass-bins, even from negative mode ion-trap MS/MS data.

Next, again mimicking our original approach, also LC-MS data by Moh et al. were extracted, charge-deconvoluted, deisotoped, and summed across the chromatographic time-range using custom code. In contrast to our original approach, however, we could not make use of the Decon2 algorithm (Decon2 was found incompatible with low-resolution ion-trap MS-data). Instead, we implemented a series of custom scripts for this purpose (i.e. MS1toPseudoMGF.pl, SugarQbits_DD.pl; Rev.Fig.6. B). From this transformed LC-MS data, precursor mass-to-intensity arrays in the range of 1000 - 4000 Da (bin-width = ± 0.5 amu) were constructed and aligned with N-glycan precursor mass-bins identified upon SNOG filtering (Rev.Fig.6. B), as described above.

For further comparison, we then aligned the results of Moh et al. with our data (Fig. Rev2.6. B). For this, the theoretical mass values of N-glycan precursors (as were provided in the supplementary tables by Moh et al.) were transformed from [M-H]⁻ to [M+H]⁺, assigned to precursor mass-bins (of bin-width = 0.5 amu) and paired with the respective summed "peak area" (as were provided in the supplementary tables by Moh et al.).

Remarkably, our data-alignment shows that the mass-bins, which were positively identified by the implementation of our computational approach, do cover most of the highly abundant N-glycan signals, which were also manually assigned by Moh, et al.. Less abundant N-glycan precursors were not picked up by our approach; due to the paucity of diagnostic fragment ions in the respective raw data.

Furthermore, our alignment highlights substantial differences with regard to precursor intensity values, that were extracted by the two analyses (e.g. mass-bin 1465.5 amu; Rev.Fig.6. B) vary between the two results. Close inspection of the raw LC-MS data by Moh et al. shows only up to 3 MS data points across the respective chromatographic peak (Rev.Fig.6.C), which is sub-optimal for its

precise reconstruction and subsequent calculation of reported “peak-areas” (under the curve), and suggests also a certain margin of deviation in the quantitative information provided by Moh et al..

Rev. Fig.6

Rev. Fig.4 – Showcase of our computational approach applied to negative mode LC-MS/MS ion-trap raw data on mouse brain N-glycome. A) Differential analysis of precursor mass-bins identified by our automated approach (using modified SNOG filtering parameters, green) and those originally manually annotated by Moh et al. (blue). B) Quantitative results of our automated approach (using a custom MS1 intensity extraction method, green) and “peak areas” originally manually annotated by Moh et al. (blue). Precursor mass-bin 1465.5 shows marked quantitative differences between the two analyses. C) The extracted ion chromatogram of precursors corresponding to mass-bin 1465.5 (i.e. $m/z = 731.3$, charge-state = 2-) indicates a limited number of LC-MS raw data-points across the chromatographic peak.

Taken together, our showcase confirms, that our computational approach, which comprises raw data refinement, data-aggregation into precursor mass-bins and the application of a modified SNOG-score based filtering step, effectively identifies N-glycan precursor masses even from seemingly incompatible LC-MS/MS data-sets, and that the results conform surprisingly well with those obtained from expert manual annotation.

Minor- 1. pg 23, line 639- female (f) of male (M)- relace 'of' by 'or' 2. Pg 5, line 28, remove an extra period.

We now corrected these mistakes and thank the reviewer for pointing them out to us!

Reviewer #3 (Remarks on code availability):

Github has data analysis files that could be used by the community. GlycoPost has the raw data files.

Point-by-point reply

Reviewer #1 (Remarks to the Author):

I appreciate the addition of the benchmark analysis and manual annotation, which enhances the credibility of the results.

I understand that the main contribution of this work is the development of computational methods aimed at profiling classes of N-glycans to characterize the diversity and heterogeneity of the N-glycome across various mouse tissues, rather than focusing on the specific profiling of individual N-glycans. Given the challenges in accurately determining detailed glycan structures and the lack of comprehensive data on glycan isomer analysis, I would recommend that the title and the main text be revised to more accurately reflect the scope of the study. Specifically, using the term "non-targeted N-glycome profiling" or "profiling classes of N-glycans" instead of "isomer-sensitive analysis of glycan structures" would provide a more precise description of the research and prevent overstating the findings.

After the thorough revisions, I look forward to seeing this article published.

We thank the reviewer for her/his positive assessment! In line with the reviewers' recommendations, we now changed the title of our manuscript to: "Non-targeted N-glycome profiling reveals new layers of organ-specific diversity in mice."

Reviewer #1 (Remarks on code availability):

I have already tested it during the first round of review, and it worked, so I did not retest it this time.

Reviewer #2 (Remarks to the Author):

The authors have thoroughly addressed my comments. I appreciate their efforts to compare the results from their oxonium-ion-based glycan identification algorithm on the mouse brain data. Originally, I suggested to use synthetic glycan standard to perform the testing. But given the extra work to perform the experiment, I will not insist to add this experiment into the current manuscript. Perhaps the authors can consider this benchmark in their future work. I still have a little bit concern about the quantification method based on spectral counts, because the spectra counts may also be indirectly dependent on the responses of glycans in mass spectrometry. The authors may want to acknowledge the limitation of this semi-quantification method.

We thank the reviewer for his/her appreciation for our benchmarking efforts. In line with the reviewers suggestion, we added a statement on the limitations of our

semi-quantitative spectral counting approach, at the end of the respective section:
“Although merely providing rough estimates on the relative abundance of specific N-glycan features, our MS/MS based N-glycome profiling approach presents an easy, fast, and robust way to automatically screen individual samples for relevant N-glycome modifications.”

Reviewer #2 (Remarks on code availability):

I did not review the new version of the code. I reviewed previously, and the code was okay.

Reviewer #3 (Remarks to the Author):

The authors improved the manuscript as requested and made an appropriate revision. I recommend the revised manuscript for publication/.

We thank the reviewer for her/his positive assessment!

Reviewer #3 (Remarks on code availability):

The code is simple and usable by the scientific community.

We thank all reviewers for their efforts and their insightful comments!